# Joint-Predictive Representations for Multi-Agent Reinforcement Learning

## Abstract

The recent advances in reinforcement learning have demonstrated the effectiveness of vision-based self-supervised learning (SSL). However, the main efforts on this direction have been paid on single-agent setting, making multi-agent reinforcement learning (MARL) lags thus far. There are two significant obstacles that prevent applying off-the-shelf SSL approaches with MARL on a partially observable multi-agent system : (a) each agent only gets a partial observation, and (b) previous SSL approaches only take consistent temporal representations into account, while ignoring the characterization that captures the interaction and fusion among agents. In this paper, we propose **M**ulti-**A**gent **Jo**int-Predictive **R**epresentations (MAJOR), a novel framework to explore self-supervised learning on cooperative MARL. Specifically, we treat the latent representations of local observations of all agents as the sequence of masked contexts of the global state, and we then learn effective representations by predicting the future latent representations for each agent with the help of the agent-level information interactions in a joint transition model. We have conducted extensive experiments on wide-range MARL environments, including both vision-based and state-based scenarios, and show that our proposed MAJOR achieves superior asymptotic performance and sample efficiency against other state-of-the-art methods.

## 1 Introduction

Representation learning has played an important role in recent developments of reinforcement learning (RL) algorithms. Especially self-supervised representation learning (SSL) has attracted more and more attention due to its success in other research fields (He et al., 2019; Devlin et al., 2018; Liu et al., 2019; Lan et al., 2019). Recently, numerous works(Srinivas et al., 2020; Zhu et al., 2022; Yarats et al., 2021; Schwarzer et al., 2021a; Yu et al., 2022) have borrowed insights from different areas and attempted to use SSL-based auxiliary tasks to learn more effective representations of RL and thus improve the empirical performance. Through augmentation for inputs, it can conduct multiple views for building SSL learning objectives, allowing the agent to improve data efficiency and generalization to obtain better task-related representations. Moreover, many proper auxiliary self-supervision priors proposed predictive representations with the help of an additional learned dynamic model, which is utilized to encourage the representations to be temporally predictive and consistent.

However, when meeting partially observable multi-agent systems, it is challenging to apply such self-supervision priors to learn compact and informative feature representations in multi-agent reinforcement learning (MARL). The critical obstacle to learning effective representations in MARL is that agents in partially observable multi-agent systems only have access to their observations, which means that other agents' behavior influences each agent's observations. As a result, independently building representation priors for each agent may be failed due to imperfect information. Furthermore, in the MARL context, it is more important to focus on the representations that embody the interaction and fusion between agents in the environment but not temporal representations for each agent. In other words, it is necessary to learn the representations that can take the other agents into account.

In this work, we propose a novel representation learning framework for MARL, named **M**ulti-**A**gent **Jo**int-Predictive **R**epresentations (MAJOR), which trains better representations for MARL by forc-

ing representations to be temporally predictive and consistent for all agents at the same timestep. We posit that in one timestep, latent representations of local observations of all agents can be treated as the sequence of masked contexts of the global state so that we can predict the future latent representations for each agent with the representations of corresponding actions. Accordingly, we construct a joint transition model as the bridge to connect all agents and implement the interaction of their individual information. The joint transition model treats the encoded representations of individual observations and actions as a sequence and attempts to predict future representations in latent space. Meanwhile, we can also get another view of the subsequent timestep representations by feeding sampled observations into the encoder. In this way, we build the SSL objective by enforcing consistency across different perspectives of each observation. Besides, our proposed framework is a plug-and-play module for almost common-used MARL methods.

Additionally, to maximize its power, we implement an instantiation of MAJOR on the basis of the recently proposed MARL algorithm, named Multi-Agent Transformer (MAT, (Wen et al., 2022)), which solves MARL issues via a sequential updating mechanism and an encoder-decoder architecture. In MAT, the encoder seeks to extract post-interaction representations from observations; the decoder then uses them to generate actions sequentially by a cross-attention mechanism. Moreover, our proposed MAJOR can employ both representations generated from the encoder and decoder, and the gradient derived from our representation learning objective can be back-propagated to both the encoder and decoder.

To evaluate our proposed algorithm, we construct extensive experiments on several common-used cooperative MARL benchmarks, including vision- and state-based environments in discrete and continuous scenarios against current state-of-the-art baselines such as HAPPO, MAPPO, and MAT. Results demonstrate its state-of-the-art performance across all tested tasks.

## 2 BACKGROUND

### 2.1 DEC-POMDP

Cooperative MARL problems are often modeled by decentralized Partially Observable Markov Decision Processes (Dec-POMDPs, (Oliehoek & Amato, 2016)) $(\mathcal{N}, \mathcal{S}, \{\mathcal{A}_i\}, \mathcal{T}, R, \Omega, \mathcal{O}, \gamma)$. Here, $\mathcal{N} = 1, \ldots, n$ is the set of agents, $\mathcal{S}$ is a set of states, $\mathcal{A} = \times_i \mathcal{A}_i$ is the set of joint actions, $\mathcal{T}$ is a set of conditional transition probabilities between states, $\mathcal{T}(s, \boldsymbol{a}, s') = P(s' \mid s, \boldsymbol{a})$, $R : \mathcal{S} \times \mathcal{A} \to \mathbb{R}$ is the reward function, $\mathcal{O} = \times_i \mathcal{O}_i$ is a set of observations for agent $i$, $\Omega$ is a set of conditional observation probabilities $\Omega(s', \boldsymbol{a}, \boldsymbol{o}) = P(\boldsymbol{o} \mid s', \boldsymbol{a})$, and $\gamma \in [0, 1]$ is the discount factor. At each time step, each agent takes an action $a_i$, and the state is updated based on the transition function (using the current state and the joint action). Each agent observes an observation based on the observation function $\Omega(s', \boldsymbol{a}, \boldsymbol{o})$ (using the next state and the joint action) and a reward is generated for the entire team based on the reward function $R(s, \boldsymbol{a})$. The goal is to maximize the expected cumulative reward over a finite or infinite number of steps.

### 2.2 MULTI-AGENT TRANSFORMER

Multi-Agent Transformer (MAT, Wen et al. (2022)) effectively casts cooperative MARL into Sequential Modeling (SM) problems wherein the task is to map the observation sequence of agents to the optimal action sequence of agents. Its sequential update scheme is built on the Multi-Agent Advantage Decomposition Theorem (Kuba et al., 2021) and Heterogeneous-Agent Proximal Policy Optimization (HAPPO, Kuba et al. (2022)). The lemma provides an intuition guiding the choice of incrementally improving actions, and HAPPO fully leverages the lemma to implement multi-agent trust-region learning with a monotonic improvement guarantee. Unfortunately, HAPPO requests the sequential update scheme in the permutation for agents' orders, meaning that HAPPO cannot be run in parallel. To address the drawback of HAPPO, MAT produces Transformer-based implementation for multi-agent trust-region learning.

Concretely, MAT maintains an encoder-decoder structure where the encoder maps an input sequence of tokens to latent representations. Then the decoder generates a sequence of desired outputs in an auto-regressive manner wherein, at each step of inference, the Transformer takes all previously generated tokens as the input. In other words, MAT treats a team of agents as a sequence, thus

implementing the sequence-modeling paradigm for MARL. The encoder $\phi$ takes a sequence of observations $\boldsymbol{o} \triangleq (o^{i_1}, \ldots, o^{i_n})$ in arbitrary order [1] and passes them through $L$ computational blocks. Each of these blocks has a self-attention mechanism(Vaswani et al., 2017) and a multi-layer perceptron (MLP), as well as residual connections to prevent gradient vanishing and network degradation as depth increases. Thus we can obtain the encoding of the observations as $\hat{\boldsymbol{o}}$ containing interrelationships among agents. Feeding the representations into the value head (an MLP), denoted as $f_\phi$, will get value estimations. The encoder's learning objective is to minimize the individual version of empirical Bellman error by:

$$\mathcal{L}^{\text{MAT}_{encoder}}_{\phi, f_\phi}(\boldsymbol{o}_t) = \frac{1}{Tn} \sum_{m=1}^{n} \sum_{t=0}^{T-1} \left[ R(s, \boldsymbol{a}_t) + \gamma V_{\phi', f'_\phi}\left(\hat{o}^{i_m}_{t+1}\right) - V_{\phi, f_\phi}\left(\hat{o}^{i_m}_t\right) \right]^2 \tag{1}$$

where $\phi'$ is the target network, which is nondifferentiable and updated every few epochs. Meanwhile, the decoder $\theta$ passes the embedding joint action to a sequence of decoding blocks. Crucially, the decoding block replaces the encoder's self-attention mechanism with a masked self-attention mechanism; i.e., the attention of the action to be generated in the current step is computed only among previously computed agents' actions. The output of the last decoder block is a sequence of representations of the joint actions. The same as the value head, this is fed into a policy head (also an MLP), denoted as $f_\theta$, which outputs the policy $\pi^{i_m}_\theta\left(\text{a}^{i_m} \mid \hat{\boldsymbol{o}}^{i_{1:n}}, \boldsymbol{a}^{i_{1:m-1}}\right)$. Besides, the decoder's learning objective is to minimize the clipping PPO objective proposed in HAPPO of

$$\mathcal{L}^{\text{MAT}_{Decoder}}_{\theta, f_\theta}(\boldsymbol{o}_t, \boldsymbol{a}_t) = -\frac{1}{Tn} \sum_{m=1}^{n} \sum_{t=0}^{T-1} \min\left(\text{r}^{i_m}_t(\theta)\hat{A}_t, \text{clip}\left(\text{r}^{i_m}_t(\theta), 1 \pm \epsilon\right)\hat{A}_t\right) \tag{2}$$

where $\mathbf{r}^{i_m}_t(\theta) = \frac{\pi^{i_m}_\theta\left(a^{i_m}_t | \hat{\boldsymbol{o}}^{i_{1:n}}_t, \hat{\boldsymbol{a}}^{i_{1:m-1}}_t\right)}{\pi^{i_m}_{\theta_{\text{old}}}\left(a^{i_m}_t | \hat{\boldsymbol{o}}^{i_{1:n}}_t, \hat{\boldsymbol{a}}^{i_{1:m-1}}_t\right)}$ and $\hat{A}_t$ is an estimation of the joint advantage function, e.g. GAE(Schulman et al., 2015).

## 3 OUR METHOD

### 3.1 CONNECTION BETWEEN JOINT TRANSITION MODEL AND SSL IN MARL

To encourage the observations and actions representations to integrate agent-level relationships and interactions, we exploit a multi-agent world model in latent space as the bridge to connect both observation representations and action representations of overall agents. Naturally, as described in the formulation of Dec-POMDP, one can feed the global state with joint actions into the world model $\mathcal{T}$ to predict the next state, i.e. implement $\mathcal{T}(s, \boldsymbol{a}, s') = P(s' \mid s, \boldsymbol{a})$. In this work, we learn a transformer-based joint transition model $\hat{\mathcal{T}}$ as an approximate version of the multi-agent world model, whose inputs are joint observation representations and joint action representations, and its outputs are the joint observation representations in the next timestep.

On the one hand, to achieve the goal of implementing information fusion among agents, it is better to use individual observations rather than a global state as input. As a result, it will enforce the learned joint transition model to (a) reconstruct the global state from individual observations, (b) predict the future state of the next timestep, and (c) implement the observation mapping functions for each agent. In this way, the joint transition model must infer the influences caused by others and try to integrate all the imperfect information. As a result, executing consistency across different views of individual observations can lead to better representations generated from encoder networks.

On the other hand, it is necessary to emphasize that we **treat the individual observations as a sequence of masked contexts of global state** in the joint transition model. Based on the definition of Dec-POMDP, observations are the outputs of the mapping function; in other orders, individual observations can be seen as multiple views of the global state. This property revealed by Dec-POMDP's formulation in cooperative MARL inspires us to propose a self-supervised prior tailored MARL. We use the joint transition model as a Masked Language Model (MLM, Devlin et al. (2018); Lewis et al. (2019)) under core components in MARL, which is general in all MARL settings.

---

[1]Because the algorithm makes no assumptions about the order of the agents and don't change the original order, we subsequently omit the superscript about the agents' order for the sake of simplicity of presentation.

### 3.2 MULTI-AGENT JOINT-PREDICTIVE REPRESENTATIONS

Multi-Agent Joint-Predictive Representations(MAJOR) is an auxiliary objective to promote representation learning in MARL and is generally applicable to different MARL algorithms, e.g., MAT and MAPPO. The core process in the joint transition model of MAJOR is to **implicitly reconstruct the global state and then infer the future observation representation of each agent**. This mechanism enables the better use of agent-cross information when learning observation and action representations, further enhancing the understanding of MARL agents for individual messages. The overall framework of MAJOR is shown in Figure 1. We will introduce the components shown in the framework in the following subsections.

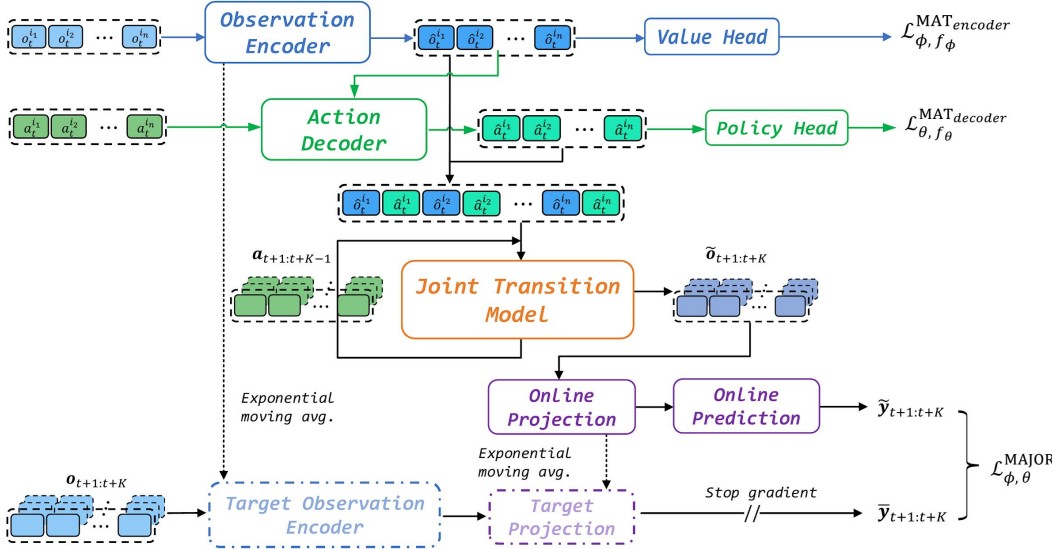

Figure 1: An illustration of the full MAJOR method. representations from the online observation encoder are used in the action decoder, value head, and policy head. They will also be utilized for the prediction of future representations from the target observation encoder via the joint transition model. The target observation encoder and projection head are defined as an exponential moving average of their online version and are not updated via gradient descent. For brevity, we illustrate the iterative $K$-steps results described in Equation 6 for the joint transition model totally in one loop.

**Encoding observations and actions:** We use MAT's encoder $\phi$ and decoder $\theta$, excluding the final MLP individual, as the *online* observation encoder and action encoder, respectively. Taking the observation sequence of arbitrary order $o_t$ as input, the online observation encoder applies a self-attention mechanism and obtains post-interaction representations of agents, as $\hat{o}_t$. Similarly to the online observation encoder, the online action encoder accepts both the origin action sequence $a_t$ and observation representations $o_t^{i_{1:n}}$ and the output action representations $\hat{a}_t$ through the cross-attention mechanism. We employ these representations with a goal that motivates them to forecast future observation representations up to a given temporal offset $K$, iteratively. Following prior work (Schwarzer et al., 2021a; Zhu et al., 2022; Yu et al., 2021b; 2022), we utilize another observation encoder for the encoding of original observations. This *target* encoder has the same architecture as the online observation encoder, and its parameters are an exponential moving average (EMA) of the online observation encoder parameters. Denoting the target observation encoder as $\bar{\phi}$ and the momentum coefficient as $\tau \in [0, 1)$, the update scheme of the target observation encoder is:

$$\bar{\theta} \leftarrow \tau\bar{\theta} + (1 - \tau)\theta \tag{3}$$

**Joint Transition Model:** We construct the predictive version of future observation representations using a transformer-based joint transition model $\hat{\mathcal{T}}$. The architecture of the joint transition model is similar to the online encoder, but (a) contains $L'$ Multi-Head Self-Attention (MHSA) layers without masks and (b) takes the combined sequence of observations and action representations as input tokens and then only outputs the sequence of observation representations of the subsequent timestep.

We obtain the $t$-th observation and action representations by feeding the origin observation sequence and the action sequence into the online observation encoder and the action encoder. Then the input tokens of the latent joint transition model can be mathematically represented as :

$$\boldsymbol{x} = [\hat{o}_t^{i_1}, \hat{a}_t^{i_1}, \ldots, \hat{o}_t^{i_n}, \hat{a}_t^{i_n}] \tag{4}$$

The process of passing the token sequence through the $l$ th layer of the joint transition model can be mathematically described as:

$$\boldsymbol{h}^l = \text{MHSA}\left(\text{LN}\left(\boldsymbol{x}^l\right)\right) + \boldsymbol{x}^l, \quad \boldsymbol{x}^{l+1} = \text{FFN}\left(\text{LN}\left(\boldsymbol{h}^l\right)\right) + \boldsymbol{h}^l \tag{5}$$

Here, LN and FFN denote the LayerNorm and the Feed-Forward Network mentioned in (Vaswani et al., 2017). Note that if the permutation order is known, one can also add agent ids' embedding and positional embedding on $\boldsymbol{x}$. And we only select the odd elements of the output tokens of the joint transition model as the corresponding predictive results for the latent future representations inferred from previous observation and action representations. Furthermore, in the $k$-th step of generating future representations where $k = 2, \ldots, K$, we use the joint transition model output representations instead of the online observation encoders as the input latent observation tokens. The process mentioned above can be denoted as

$$
\begin{aligned}
&\tilde{\boldsymbol{o}}_{t+1} \triangleq \hat{\mathcal{T}}\left(\hat{\boldsymbol{o}}_t, \hat{\boldsymbol{a}}_t\right), \text{ where } \hat{\boldsymbol{o}}_t \triangleq \phi\left(\boldsymbol{o}_t\right), \text{ and } \hat{\boldsymbol{a}}_t \triangleq \theta\left(\hat{\boldsymbol{o}}_t, \boldsymbol{a}_t\right) \\
&\tilde{\boldsymbol{o}}_{t+k} \triangleq \hat{\mathcal{T}}\left(\tilde{\boldsymbol{o}}_{t+k-1}, \tilde{\boldsymbol{a}}_{t+k-1}\right), \text{ where } \tilde{\boldsymbol{a}}_{t+k-1} \triangleq \theta\left(\tilde{\boldsymbol{o}}_{t+k-1}, \boldsymbol{a}_{t+k-1}\right), \forall k = 2, \ldots, K
\end{aligned} \tag{6}
$$

The joint transition model and prediction loss operate in the latent space, thus avoiding pixel-based reconstruction objectives and making MAJOR robust for vision-based and state-based MARL settings. In the following, we elaborate on the predictive loss between the prediction results and the corresponding targets.

**Prediction Loss:** Motivated by the success of BYOL (Grill et al., 2020) in SSL and sample-efficient RL(Schwarzer et al., 2021a; Yu et al., 2021b; 2022), we compute the future prediction loss of MAJOR by calculating the cosine similarities between the predicted and observed representations. Concretely, from the outputs of the joint transition model, i.e. the sequence of observation representations set $\tilde{\boldsymbol{o}}_{t+1:t+K}$, we use a projection head $g$ and a prediction head $q$ to obtain the final sequence of predictions result in $\tilde{\boldsymbol{y}}_{t+1:t+K} \triangleq q(g(\tilde{\boldsymbol{o}}_{t+1:t+K}))$. Then we utilize a target projection head $\bar{g}$ (i.e. follows the same EMA update strategy in the target observation encoder) to process the encoded results of original observations, which is denoted as $\bar{\boldsymbol{y}}_{t+1:t+K} \triangleq \bar{g}(\bar{\boldsymbol{o}}_{t+1:t+K})$ where $\bar{\boldsymbol{o}}_{t+1:t+K} \triangleq \bar{\phi}(\boldsymbol{o}_{t+1:t+K})$. Here, we apply a stop-gradient operation as illustrated in Figure 1 to avoid model collapse, following BYOL. Finally, MAJOR's objective is to enforce the final prediction result in $\tilde{\boldsymbol{y}}_{t+1:t+K}$ to be as close to its corresponding target $\bar{\boldsymbol{y}}_{t+1:t+K}$. And we construct the following cosine similarities between the normalized predictions and the target projections overall agents and the offset timesteps:

$$\mathcal{L}_{\phi,\theta}^{\text{MAJOR}}(\boldsymbol{o}_{t:t+K}, \boldsymbol{a}_{t:t+K-1}) = -\frac{1}{Kn} \sum_{k=1}^{K} \sum_{i=1}^{n} \left(\frac{\tilde{y}_{t+k}^i}{\|\tilde{y}_{t+k}^i\|_2}\right)^\top \left(\frac{\bar{y}_{t+k}^i}{\|\bar{y}_{t+k}^i\|_2}\right) \tag{7}$$

**Total learning objective:** The proposed MAJOR is an auxiliary task that is optimized in conjunction with MAT. Therefore, the overall loss function is:

$$\mathcal{L}_{total} = \mathcal{L}_{\phi,f_\phi}^{\text{MAT}_{encoder}} + \mathcal{L}_{\theta,f_\theta}^{\text{MAT}_{decoder}} + \lambda \mathcal{L}_{\phi,\theta}^{\text{MAJOR}} \tag{8}$$

where $\mathcal{L}_{\phi,f_\phi}^{\text{MAT}_{encoder}}$, $\mathcal{L}_{\theta,f_\theta}^{\text{MAT}_{decoder}}$ and $\mathcal{L}_{\phi,\theta}^{\text{MAJOR}}$ are the MAT losses mentioned above and our proposed joint-predictive representations learning objective, respectively. $\lambda$ is a hyperparameter for balancing the items. It is worth noting that, unlike other suggested SSL algorithms in CV and RL, MAJOR can be employed with or without data augmentation, especially in situations where data augmentation is unavailable or counterproductive. Moreover, MAJOR mainly focuses on capturing the relationships among agents via the joint transition model. The proposed framework can also be transferred to other MARL algorithms that follow the centralized training decentralized execution (CTDE) paradigm, such as MAPPO(Yu et al., 2021a)/HAPPO(Kuba et al., 2022) and QMIX(Rashid et al., 2018)/QPLEX(Wang et al., 2020), etc. Here, we provide the pseudocode of MAJOR in Algorithm 1.

---

**Algorithm 1** Multi-Agent Joint-Predictive Representations

---

1: **Input:** Stepsize $\alpha$, number of agents $n$, episodes $K$, steps per episode $T$.
2: **Initialize:** Observation encoder $\phi$, Action decoder $\theta$, Value head $f_\phi$, Policy head $f_\theta$, Replay buffer $\mathcal{B}$, Joint transition model $\hat{\mathcal{T}}$, Online projection head $g$, Online prediction head $q$, Target observation Encoder $\bar{\phi}$, Target projection head $\bar{g}$.
3: **while** Training **do**
4:     Sample a minibatch $(\boldsymbol{o}_t, \boldsymbol{a}_t, r_t, \boldsymbol{o}_{t+1}) \sim \mathcal{B}$.
5:     Calculate $\mathcal{L}^{\text{MAT}_{encoder}}_{\phi, f_\phi}(\boldsymbol{o}_t)$ with Equation (1).
6:     Calculate $\mathcal{L}^{\text{MAT}_{Decoder}}_{\theta, f_\theta}(\hat{\boldsymbol{o}}_t, \boldsymbol{a}_t)$ with Equation (2).
7:     Sample another sequential minibatch $(\boldsymbol{o}_{t:t+K}, \boldsymbol{a}_{t:t+K-1}) \sim \mathcal{B}$.
8:     Get projection/prediction representations $\tilde{\boldsymbol{y}}_{t+1:t+K}$ and $\hat{\boldsymbol{y}}_{t+1:t+K}$ with Equation (6).
9:     Calculate $\mathcal{L}^{\text{MAJOR}}_{\phi, \theta}(\boldsymbol{o}_{t:t+K}, \boldsymbol{a}_{t:t+K-1})$ with Equation (7)
10:    Update the encoder/decoder, value/policy head, joint transition model, and online projection/prediction head by minimizing $\mathcal{L}_{total}$ with gradient descent according to Equation (8).
11:    Update target observation encoder and target projection head with Equation (3)
12: **end while**

---

## 3.3 Implement details for MAJOR

In practice, we define a hyperparameter $H$ to control the update frequency of MAJOR's loss, w.r.t. MAT's loss. In other words, we update MAJOR's learning objective every certain number of iterations of the original MAT optimization strategy. Concretely, we sample a unique batch of $B'$ samples from the trajectories collected using the latest policy. Each sample consists of a temporal set of observations and the corresponding actions, that is, $(\boldsymbol{o}_{t:t+K}, \boldsymbol{a}_{t:t+K})$. For the projection and prediction head, we do not use BatchNorm Layer and replace ReLU with GELU activation units, which is different from what BYOL did. In addition, our joint transition model's architecture is the same as the encoder used in MAT. In vision-based settings, we use three convolutional layers with a ReLU layer after each convolutional layer, which is the same as DQN's, as the feature extractor in all algorithms. The full hyperparameters of MAJOR can be found in Appendix F.

## 4 Related Work

### 4.1 Overview of Representation Learning and Self-Supervised Learning

Self-supervised learning empowers us to exploit a variety of labels that come with the data for free. With self-supervised learning, we can utilize inexpensive unlabeled data and establish the learning objectives properly from designed pretexts to gain supervision from the data itself. SSL has been developed in CV and NLP areas and can be divided into the various self-supervised pretexts in the literature into four broad families(Ericsson et al., 2022): *Masked Prediction*, *Transformation Prediction*, *Instance Discrimination*, and *Clustering*. (1) Masked Prediction methods (Mikolov et al., 2013; Baevski et al., 2020; Pathak et al., 2016; Hu et al., 2020) mask a portion of word tokens or image pixels from the input sentence or image and train the model to predict the masked components to obtain effective representations. (2) Transformation Prediction methods (Gidaris et al., 2018; Sarkar & Etemad, 2020; Xu et al., 2019) apply a transformation that maps from canonical views to alternative views and trains the model to predict what transformation has been applied. (3)Instance Discrimination methods (Velickovic et al., 2019; Chen et al., 2020; He et al., 2019; Tian et al., 2020) apply some transformation process in one instance to obtain multiple views of it and attempt to formalize the contrastive instance discrimination. (4) And Clustering methods (Caron et al., 2018; 2020; Zhan et al., 2020; Alwassel et al., 2020) focus on dividing the training data into several groups with high intra-group similarity and low inter-group similarity. We recommend readers read (Ericsson et al., 2022) to get more information.

### 4.2 Self-Supervised Learning and Reinforcement Learning

There exist substantial works taking advantage of SSL techniques to promote representation learning in RL. A popular approach is to jointly learn policy learning objectives and auxiliary objectives.

As for constructing auxiliary SSL objectives, the primary way is to build multiple views of the same input through masked-latent reconstruction or dynamic models with augmentations. For instance, (Srinivas et al., 2020; Zhu et al., 2022) tried to extract high-level features from raw pixels using contrastive learning and performed off-policy control on top of the extracted features. (Schwarzer et al., 2021a; Yu et al., 2021b; 2022; Zhang et al., 2021) leverage a dynamic model to obtain a predicted version of the subsequent observation and then use contrastive learning to enforce consistency between the raw future observation and the prediction version of it in latent space. Another alternative way of obtaining good representations is to pre-train the observation encoder to learn effective representations before policy learning(Yarats et al., 2021; Stooke et al., 2021; Schwarzer et al., 2021b; Yang & Nachum, 2021; Campos et al., 2021).

### 4.3 MULTI-AGENT REINFORCEMENT LEARNING AND REPRESENTATION LEARNING

Recent works (Yeh et al., 2019; Sun et al., 2019; Zhan et al., 2018; Yuan et al., 2021) study the representation learning for developing models that predict trajectories over all agents at once but not under the MARL context. And as far as we know, only a few works consider promoting representation in the MARL context. The most similar works are (Shang et al., 2021) and (Zhang et al., 2022). (Shang et al., 2021) task each agent to predict its future location, arriving at an agent-centric predictive objective to be combined in their proposed agent-centric attention module in the football game. (Zhang et al., 2022) is a model-based MARL method that proposed a graph-assisted predictive state representation learning framework that leverages the agent connectivity graphs to aggregate local representations computed by each agent. Note that the SSL prior proposed in (Shang et al., 2021) only be used in football-like environments and is not flexible. Additionally, our method aims to build a general plugin for model-free MARL approaches so that model-based MARL methods are not directly comparable to our method. We focus on the auxiliary task line in this work.

## 5 EXPERIMENTS

We consider a series of MARL benchmarks to evaluate MAJOR, including vision-based and state-based MARL benchmarks. All hyperparameters settings can be found in Appendix G.

### 5.1 SETUP

**Vision-based MARL Environments.** Representation Learning and SSL techniques show their strength in vision-based RL environments, such as DMControl and Atari. To evaluate whether our proposed MAJOR is also powerful in such vision-based MARL settings, we run it on three physics-based cooperative tasks in Multi-Agent Quadcopter Control(dubbed MAQC) (Panerati et al., 2021). MAQC is an open-source, OpenAI Gym-like multi-quadcopter simulator that provides vision-based observations and multi-agent controlling interfaces. Observations include video frames from the perspective of each drone (toward the positive direction of the local x-axis) for the RGB ($\in \mathbb{R}^{64 \times 48 \times 4}$), depth, and segmentation ($\in \mathbb{R}^{64 \times 48 \times 1}$) views. The action of drones is continuous velocity and the magnitude of the velocity.

Here, we briefly introduce the two scenarios, named *Flock* and *LeaderFollower* in MAQC. Denote $i$-th agent's xyz coordinates as $\mathbf{x} = (x, y, z)$, individual reward as $r_i$, team reward is $R = \sum_{i=1}^{n} r_i$, and then: (i) In *Flock*, the first agent should keep its position with a predefined location (e.g., $\mathbf{p}$) as close as possible, and $i$-th agents($i > 1$) need to track $(i-1)$-th agent's latitude, i.e., $r_1 = -\|\mathbf{p} - \mathbf{x}_1\|_2^2$, $r_i = -(y_i - y_{i-1})^2 \; \forall i = 2, \ldots, n$ (ii) The goal of *LeaderFollower* in MAQC is to train the *follower* drones to track the *leader* drone, and the leader drone needs to keep its position with a predefined position as close as possible, i.e., $r_1 = -\|\mathbf{p} - \mathbf{x}_1\|_2^2$, $r_i = -\frac{1}{n}(z_i - z_1)^2 \; \forall i = 2, \ldots, n$. An overview of MAQC is shown in Figure 2. Note that in our experiments we only use the RGB information provided by the simulator.

**State-based MARL Environments.** Since our method can be used not only in vision-based but also in state-based scenarios in MARL environments, we also test our method in state-based cooperative MARL scenarios. Our evaluation includes StarCraftII Multi-Agent Challenge (SMAC) (Samvelyan et al., 2019), Multi-Agent MuJoCo (de Witt et al., 2020), and Google Research Football (Kurach et al., 2020), in which the baselines compared show the current state-of-the-art performance.

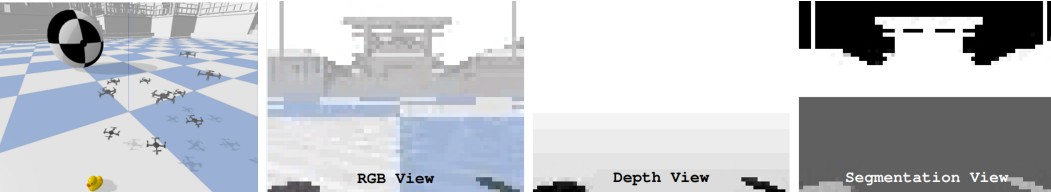

Figure 2: Overview of Multi-Agent Quadcopter Control environment and the observation for the quadrotor. A quadrotor is (i) an easy-to-understand mobile robot platform whose (ii) control can be framed as a continuous states and actions problem but, beyond 1-dimension, (iii) it adds the complexity that many candidate policies lead to unrecoverable states, violating the assumption of the existence of a stationary state distribution on the entailed Markov chain.

**Baselines and evaluations.** We select MAPPO, HAPPO, and MAT as the compared baselines because they are general to both continuous and discrete MARL environments. Moreover, they have achieved state-of-the-art performance on common-used MARL benchmarks. We keep the hyperparameters proposed in their papers for a fair comparison of all algorithms. We run five random seeds to get the mean and standard deviation of the evaluation metrics. We calculate the winning rate in SMAC and the playing scores in Google Football. For other environments, we use episode rewards as the evaluation metric. Additionally, we only select three super-hard difficulty scenarios in the SMAC environment. And we play four drones in the MAQC simulator to test the performance of candidates to be compared. As for Multi-Agent MuJoCo, we enforce agents can only observe the state of their joints and bodies but not their immediate neighbor's joints and bodies.

## 5.2 RESULTS AND ANALYSIS

**Multi-Agent Quadcopter Control.** Figure 3 shows the performance of MAJOR, compared with strong baselines in the MAQC environment. The three tasks require drones (agents) to learn to fly to a specific goal and keep a lower distance from others by controlling the velocity vector. The results in Flock and LeaderFollwer demonstrates the strength of MAJOR since it conducts the joint transition model and implicitly captures the relationships of euclidean distance via the learned joint transition model so that for the goal of these two scenarios, being closed with other drones, MAJOR can inject more helpful information into networks thus bring significant improvement w.r.t other algorithms. Moreover, the results also illustrate that it is not enough to learn effective representations from high-dimensional image-based observations only through policy optimization. So there are still many challenges in applying MARL algorithms in vision-based environments. And we firmly believe that our work will bring some inspiration for vision-based MARL studies.

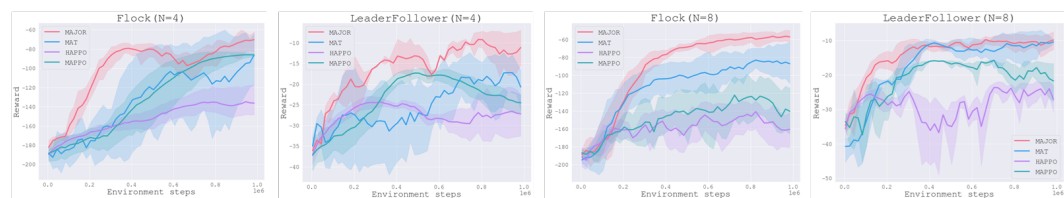

Figure 3: Results on Multi-Agent Quadcopter Control. MAJOR outperforms all prior methods on two of three tasks, and all methods cannot work on the Meetup.

**SMAC, MA MuJoCo and Google Football.** We compare MAJOR with the SOTA MARL methods for typical state-based cooperative MARL benchmarks. According to Figure 4b, Figure 4a, and Figure 4c, MAJOR also shows superior performance, which means that even though the observations are non-visual signals, MAJOR still embodies the interaction and fusion between agents through the joint transition model in the latent space. This can be significantly shown in Multi-Agent MuJoCo. To cooperate, the robot agents must attempt to infer information from other joints and bodies. Other compared methods only considered the feature encoding process from the perspective of policy optimization. Still, MAJOR can take into account other agents' positions and velocity by the extra self-supervised representation learning task, which makes it natural to achieve better performance.

Besides, our algorithm can also improve MAT's sample efficiency in Google Football and SMAC benchmarks.

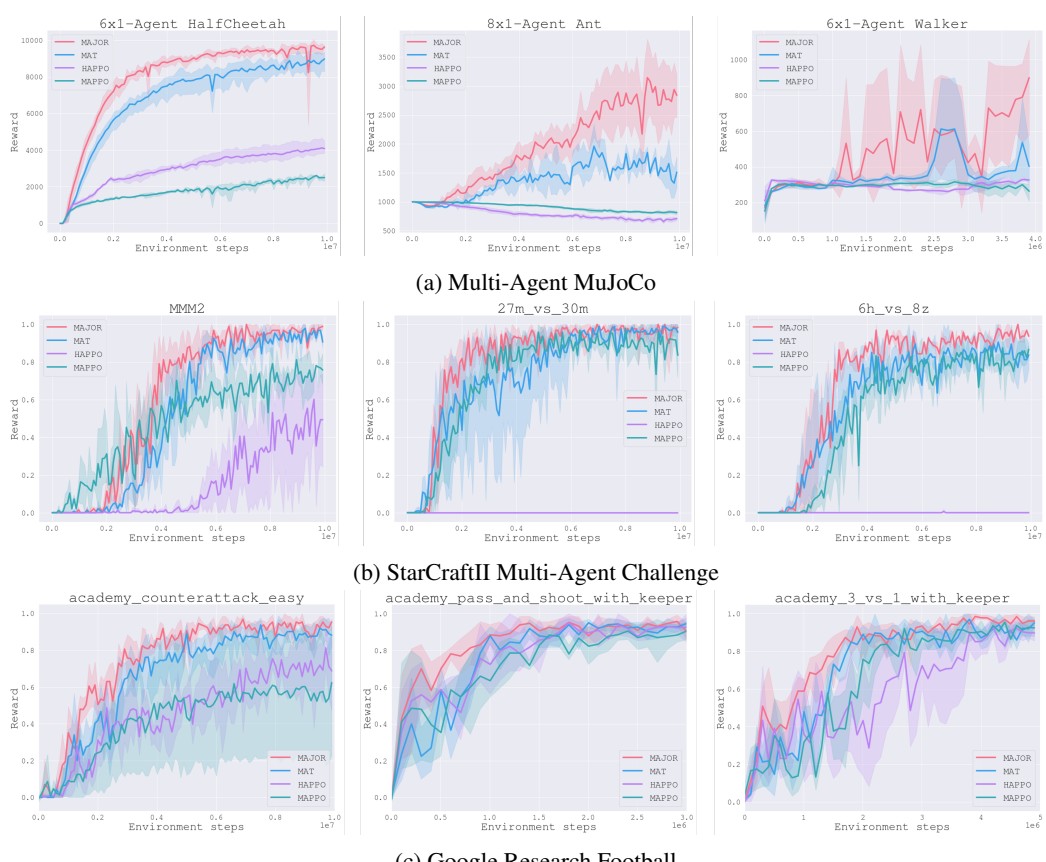

(a) Multi-Agent MuJoCo

(b) StarCraftII Multi-Agent Challenge

(c) Google Research Football

Figure 4: Performance for MAJOR, MAT, HAPPO, and MAPPO on (a) Multi-Agent MuJoCo, (b) StarCraftII Multi-Agent Challenge, and (c) Google Research Football. MAT outperforms compared current state-of-the-art baselines in common cooperative MARL benchmarks.

## 6  CONCLUSION

In this paper, we introduce Multi-Agent Joint-Predictive Representations(MAJOR), a self-supervised representation learning algorithm designed to improve the data efficiency of MARL algorithms. MAJOR treats the individual observations as a masked sequence and learns the representations that are jointly temporally predictive and consistent across different views overall agents, by implicitly reconstructing the global state and directly predicting representations of observations produced by a joint transition model and a target encoder. Experimental results on both vision-based and state-based cooperative MARL benchmarks(i.e. Multi-agent Quadcopter Control, SMAC, Multi-agent MuJoCo, and Google Football) show MAJOR's state-of-the-art performance. Based on the established connection between MARL and SSL, in the future, we will consider incorporating cycle-consistent predictive or other powerful SSL priors into MAJOR and combining offline representation pretraining with task-specific finetuning in the paradigm.

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

# A FURTHER DISCUSSION ON MAJOR

## A.1 PERFORMANCE OF MAJOR

It is worth noting that the state-based observations in these environments usually contain the key attributes demonstrating the position, velocity, and hp of the agents, which means that there are no irrelevance or noisy messages in the vector. Thus, the neural network can easily capture useful information, and selected baselines will achieve strong performance close to the upper bound with the help of a powerful policy optimization mechanism. So our proposed representation learning task can only take slight improvements w.r.t the original baseline. In other words, for state-based observations, their representations are more effective than vision-based observations through the learning process of MARL methods.

However, in contrast to low-dimensional state-based vectors, learning representations from high-dimensional image-based observations only through policy optimization are more complicated. And our experimental results of MAQC show a significant improvement compared with baselines, which provides the benefit of extra representation learning in current MARL methods for vision-based environments.

In this work, we mainly focus on extending SSL for MARL to improve efficiency, especially in vision-based multi-agent environments. And the results of wide-range benchmarks can demonstrate the generalization of our proposed method for both vision- and state-based scenarios.

## A.2 MOTIVATION OF DESIGNING MAJOR FOR MARL

Firstly, we would like to emphasize that the goal of incorporating SSL in the single-agent RL method is to learn more effective representations when fixing the encoder and the number of dimensions of latent space, either visual inputs or state-based inputs. The more informative and rich representations contained in a latent variable, the more effective it is to perform policy optimization. However, in practice, vision-based inputs usually include many noisy and irrelevant components rather than state-based inputs, which brings more obstacles to capturing informative features. Meanwhile, state-based inputs are often designed with critical elements that reflect the physical attributes of the agent with only a few distractions. For example, when we train the same RL algorithm (e.g., SAC) on MuJoCo, it is accessible to extract features from the physical vector(i.e., velocity and position, etc.) via MLP encoder and get high scores, but it is more challenge to use the CNN encoder to obtain good representations and achieve the same performance in DeepMind Control Suite(i.e., visual MuJoCo). Thus we believe that the benefit of applying SSL to both single- and multi-agent RL is more visible on high-dimensional visual space inputs.

Secondly, as stated in the original paper, we would like to clarify that our work is not simply applied SSL to existing MARL methods. It is worth noting that the representation learning challenge under the MARL context is more difficult than the SARL situation since local observations will be affected by the change in the ego agent's policy and other agents' behavior. As a result, simply applying SSL priors for each agent may fail due to imperfect information. Furthermore, we believe that our work, which concerned embodying the interaction and fusion among agents in the environment rather than temporal representations for each agent, is not simply to apply SSL to MARL. And our work is motivated by incorporating agent-level interactions to make representations that can take the other agents into account to be more effective. Due to the time limitation of the rebuttal, we will attempt to apply the common-used SSL technique in SARL, named SPR, on MAT to verify whether simply doing the combination will work.

Thirdly, we believe that the vision-based multi-agent system will be paid more and more attention in the future days. It is no doubt that real-world scenarios, such as 2D/3D robotics and eSports games, are almost accepting visual signals but not state-based vectors. Recently, many works are proposed in the community. Shah et al. (2018); Panerati et al. (2021) present simulators for controlling drones to visual signals collected with drone cameras. Chen et al. (2022) presents a bimanual dexterous manipulation benchmark (Bi-DexHands) according to the cognitive science literature for comprehensive reinforcement learning research. A similar job that predates Chen et al. (2022) is OpenAI's human-like robot hand Akkaya et al. (2019), which solves Rubik's Cube and whose input is generated by built-in sensors. Moreover, many robotic navigation benchmarks are presented with more complex visual inputs, such as Deitke et al. (2020; 2022); Ahn et al. (2022), etc. Meanwhile,

in eSports, the famous OpenAI Five (Berner et al., 2019) and AlphaStar (Vinyals et al., 2019) also use human-like images as inputs. These developments indicate that more and more research will be studied on multi-agent environments based on visual inputs. In other words, these 2D/3D robotics and eSports areas provide powerful platforms for researchers to solve real-world problems using MARL. And research based on these areas will also have greater potential to be applied to actual industrial scenarios. However, current research on MARL seems to be lagging behind. The observations of the common-used benchmarks(e.g. SMAC, Google Research Football, MAMuJoCo) are all based on hand-designed physical values. These environments seem too simplistic compared with the real-world applications mentioned above. Thus it is meaningful to make an effort to explore the idea of representation learning in MARL for vision-based multi-agent systems rather than toy environments. And we believe our work will be useful to the community.

### A.3 Feasibility analysis of MAJOR

Note that it may not work without any assumptions in Dec-POMDP with imperfect observation mapping functions, i.e. all agent's observations may not contain all the information in the state due to the partial observation issue. In this work, we build MAJOR on MAT, and MAT utilities the local observations to estimate the expected team return under a cooperative MARL context. The MAT's encoder with the value head is used to approximate the global value function and get a good performance, illustrating that it can recover the state and then estimate the team return from the sequence of partial observations. Thus we can assume that all agent's observations in the evaluated Dec-POMDP benchmarks can also contain all the state information, leading to our work becoming practical. At the same time, in our experimental results, we find that MAJOR loss in all experiments reduced rapidly, meaning that our method can work on common-used Dec-POMDP benchmarks without pointed concerns.

As mentioned in the related work section of the original paper, Zhang et al. (2022) takes a study of learning a dynamic model under model-based MARL in Dec-POMDP, which is similar to the pointed issue. The work proposed a graph-based version adaptation of Predictive State Representations (Littman & Sutton, 2001) to infer the predictive representations and uses the representations as the input of a policy optimization algorithm. Even though the method is not directly comparable to our model-free way with the auxiliary task, Zhang et al. (2022) has implicitly proven the feasibility of predicting future representations from individual historical observation. We recommend readers to read Zhang et al. (2022) for more information. Furthermore, we believe that adding historical trajectories of agents can address the issue as much as possible. And we will involve both temporal information and agent-level interactions simultaneously in MAT and MAJOR to make them more powerful in our future work.

### A.4 Fairness of comparison between MAT and CTDE methods

There is no doubt that MAT follows the CTCE(centralized training and centralized execution) pattern. But MAT only takes partial observations rather than a global state in the training and execution phases, while CTDE approaches must depend on the global state. However, in many vision-based scenarios, the system state is enormous compared with observations since it has to use an extensive view to cover all agents' views to reflect the system(but most of its messages may be unnecessary). Therefore, the huge-volume state may become the bottleneck of all CTDE MARL methods. Because MAJOR is more effective when applied to vision-based environments, comparing MAT with typical CTDE methods will not result in many unfair issues, respectively.

## B More Experimental Results and Analysis

### B.1 Training Curves for Loss function in MAJOR and MAT

Here we put the training curve of MAJOR loss and V loss in the four-drone Flock of MAQC and give the qualitative analysis of MAJOR. As shown in Figure 5, the MAJOR loss reduces rapidly, meaning that the MAT's encoder and the joint transition model can account for the information of other agents with the help of the designed auxiliary task. Meanwhile, as shown in Figure 6, the global value function approximation also benefits from the compact representation of observations

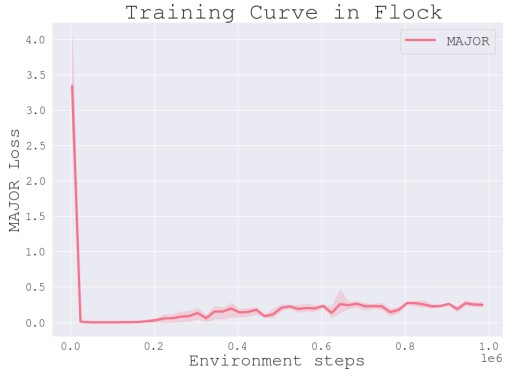

Figure 5: Training curve for MAJOR's loss.

Figure 6: Training curves for value loss.

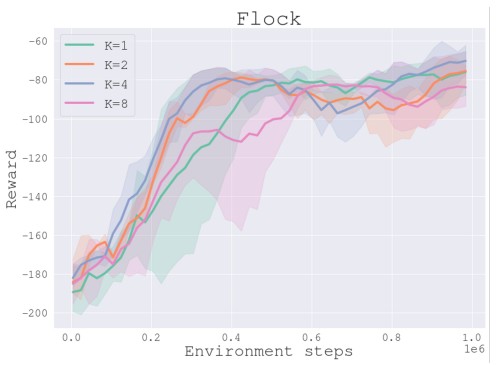 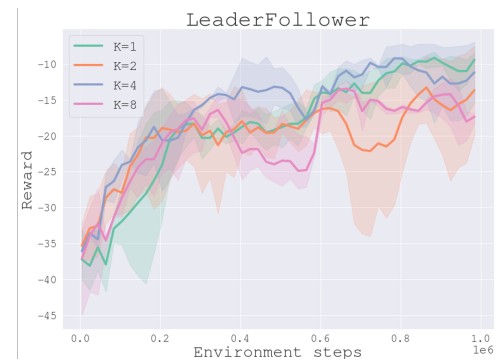

Figure 7: Ablation study of forwarding steps $K$ for the joint transition model where $H = 1$.

as training progresses since the MAJOR's value loss is lower than MAT's. The more accurately the value function fits, the better the policy optimization effect. And we would like to infer that the non-convergence policy causes the rapid overfitting issue of MAJOR's loss at the beginning of the training phase. Similar results can be found in other scenarios.

### B.2 ABLATION STUDY

Here we give more details about the ablation study. In our method, the critical hyperparameters are predefined $K$ and $H$, meaning the number of forward steps for the joint transition model and the updating frequency for MAJOR's learning objective, respectively. Figure 7 shows the results of $K$ at $\{1, 2, 4, 8\}$ in the four-drone Flock and LeaderFollower of MAQC. We find that extended dynamics modeling consistently improves performance up to roughly. A larger $K$ (e.g., 8) does not bring further improvement since the network cannot predict long-term future representations as the accumulation of single-step prediction. And Figure 8 illustrates that updating frequency can also influence the efficiency of MAT. In our opinion, lazy updating for MAJOR will make representations mismatch with the corresponding policy gradient, thus bringing a negative impact on MAT.

## C INCORPORATING MAJOR ON OTHER MARL ALGORITHMS

In this section, we start by introducing how to incorporate our proposed auxiliary task into MARL algorithms following the CTDE pattern, such as MAPPO and HAPPO. And then we will show the results of the modified version of MAPPO/HAPPO on four-drone Flock in MAQC.

As stated in Sec. 1, the benefit of combining MAT with MAJOR is that it can employ both representations generated from MAT's encoder and decoder. Thus, the gradient derived from our auxiliary learning objective can be back-propagated to both the encoder and decoder simultaneously. Contrary to MAT, common-used MARL algorithms(e.g., MAPPO, HAPPO) under the CTDE framework usu-

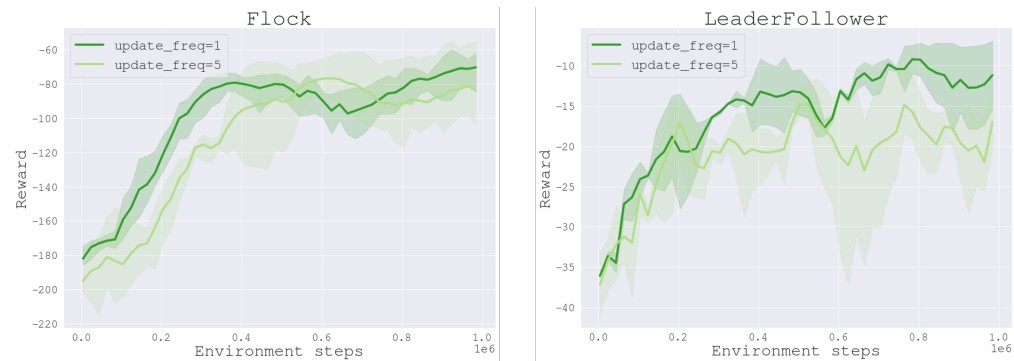

Figure 8: Ablation study of updating frequency $H$ of MAJOR's learning objective where $K = 4$.

ally take the global state as the input for the critic network, so we need to modify the pipeline of the proposed auxiliary task.

The following figure demonstrates the combined version of MAPPO. Since MAPPO's critic network does not use individual observations, we can only incorporate MAJOR in the actor and make the gradient of auxiliary loss proportionate through the actor network. We are supposing the architecture of the actor network contains more than two MLPs after the encoder. We are supposing the architecture of the actor network contains more than two MLPs after the encoder. In other words, the pipeline of the actor network is:

$o \rightarrow$ CNN/MLP(parameterized by $\theta$) $\rightarrow \hat{o} \rightarrow$ MLP(parameterized by $\phi$) $\rightarrow \hat{a} \rightarrow$ MLP(parameterized by $\omega$) $\rightarrow a$

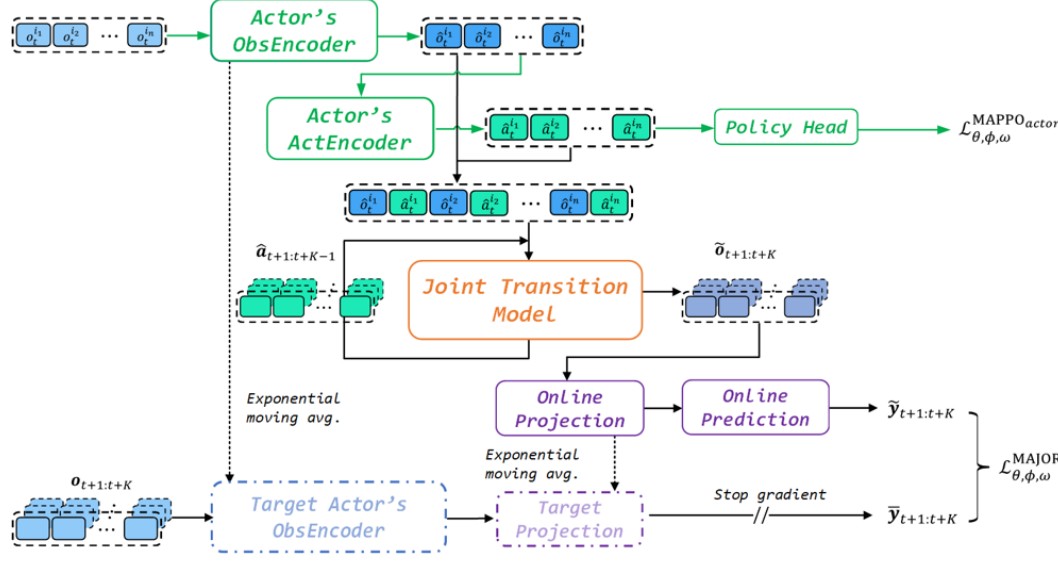

Figure 9: An illustration of incorporating MAJOR into MAPPO.

Thus, given the intermediate outputs of the actor network, we can obtain the observation and action representations. The subsequent pipeline is similar to Figure 1. As shown in Figure 9, the differences between MAPPO + MAJOR and MAT + MAJOR are:

- We treat MAPPO's actor network as the sequential combination of observation encoder(dubbed *ObsEncoder*) and action encoder(dubbed *ActEncoder*) followed by the policy head. Alternatively, you can use a separate network as ActEncoder outside the actor network.
- The actor's ActEncoder only takes the observations' representations as input.

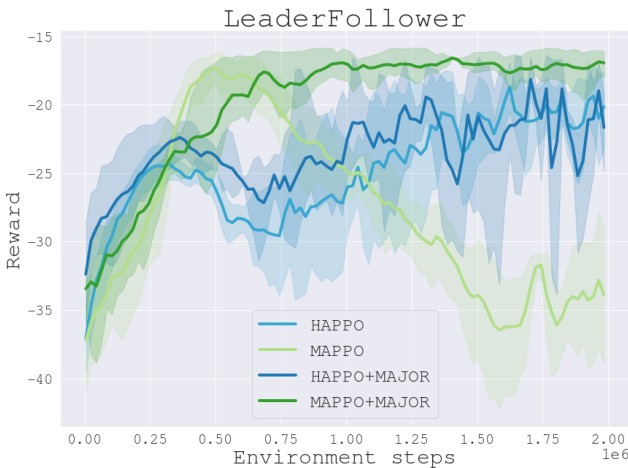

Figure 10: Results for combining MAJOR and MAPPO/HAPPO in LeaderFollow.

- We feed action representations generated from the actor's ActEncoder rather than the executed actions in sampled transitions into the joint-transition model.

Note that recently there has been some literature(Hu et al., 2021; Wang et al., 2022) utilizing Transformer models as a feature extractor for single observations, which contain a sequence of various entities. So applying MAJOR on MAPPO/HAPPO will be similar to the shown pipeline in Figure 1. In this situation, we treat the global state as a sequence of observations or elements in the system. And the joint transition model accepts representations from both actor and critic networks.

We have also evaluated the performance of the modified version of MAJOR in the four-drone LeaderFollower of MAQC. The result in Figure 10 illustrates that MAJOR can also improve both MAPPO and HAPPO. But the improvement is not significant because MAJOR can only help the actor network to learn more impact observation representations. Still, it does not benefit to critic network, which makes no improvements for value estimation.

## D EXTENDED BACKGROUND

### D.1 EXISTING METHODS IN MARL

We will now give a quick overview of two cutting-edge MARL algorithms. Both are based on Proximal Policy Optimization (PPO, Schulman et al. (2017)), an RL approach known for its simplicity and stability of performance.

**MAPPO** (Yu et al., 2021a) was the first and most straightforward technique for implementing PPO in MARL. It provides all agents with the same set of parameters and updates the shared policy based on the aggregated trajectories of the agents. In detail, at iteration $k + 1$, it optimizes the policy parameter $\theta_{k+1}$ by maximizing the clip objective of

$$\sum_{i=1}^{n} \mathbb{E}_{\mathbf{o} \sim \rho_{\boldsymbol{\pi}_{\theta_k}}, \mathbf{a} \sim \boldsymbol{\pi}_{\theta_k}} \left[ \min \left( \frac{\pi_\theta \left( \mathbf{a}^i \mid \mathbf{o} \right)}{\pi_{\theta_k} \left( \mathbf{a}^i \mid \mathbf{o} \right)} A_{\boldsymbol{\pi}_{\theta_k}}(\mathbf{o}, \mathbf{a}), \mathrm{clip} \left( \frac{\pi_\theta \left( \mathbf{a}^i \mid \mathbf{o} \right)}{\pi_{\theta_k} \left( \mathbf{a}^i \mid \mathbf{o} \right)}, 1 \pm \epsilon \right) A_{\boldsymbol{\pi}_{\theta_k}}(\mathbf{o}, \mathbf{a}) \right) \right],$$

where the clip operator (if required) trims the input value to keep it inside the interval $[1 - \epsilon, 1 + \epsilon]$. Enforcing parameter sharing, on the other hand, is analogous to imposing a restriction $\theta^i = \theta^j, \forall i, j \in \mathcal{N}$ on the joint policy space, which might result in a suboptimal conclusion that is exponentially worse (Kuba et al., 2022). This encourages the development of more principled heterogeneous-agent trust-region approaches, such as HAPPO.

**HAPPO** is one of the SOTA algorithms that completely exploits Multi-Agent Advantage Decomposition (Kuba et al., 2021) to provide multi-agent trust-region learning with monotonic improvement guarantees. During an update, the agents choose a permutation $i_{1:n}$ at random, and then, in the

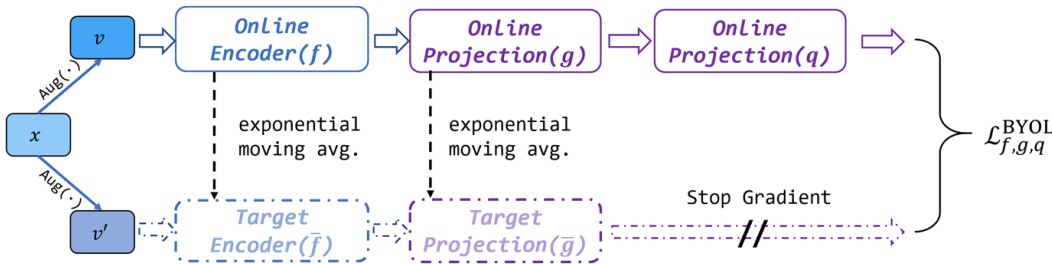

Figure 11: An illustration of the framework of BYOL.

sequence in which the permutation was chosen, each agent $i_m$ picks $\pi_{\text{new}}^{i_m} = \pi^{i_m}$ that maximizes the aim of

$$\mathbb{E}^{i_m}_{\mathbf{o} \sim \rho_{\pi_{\text{old}}}, \mathbf{a}^{i_1}_{1:m-1} \sim \pi^{i_{1:m-1}}_{\text{new}}, \mathbf{a}^i m \sim \pi^{i_m}_{\text{old}}} \left[ \min \left( \mathrm{r} \left( \pi^{i_m} \right) A^{i_{1:m}}_{\pi_{\text{old}}} \left( \boldsymbol{o}, \mathbf{a}^{i_{1:m}} \right), \mathrm{clip} \left( \mathrm{r} \left( \pi^{i_m} \right), 1 \pm \epsilon \right) A^{i_{1:m}}_{\pi_{\text{old}}} \left( \mathbf{o}, \mathbf{a}^{i_{1:m}} \right) \right) \right],$$

where $\mathrm{r}(\pi^{i_m}) = \pi^{i_m}(\mathrm{a}^{i_m}|\mathbf{o})/\pi^{i_m}_{\text{old}}(\mathrm{a}^{i_m}|\mathbf{o})$. It is worth noting that the expectation is placed over the newly-updated prior agents' policies, i.e., $\pi^{i_{1:m-1}}_{\text{new}}$; this reflects an intuitive understanding that, according to Theorem (1), the agent $i_m$ responds to its preceding agents $i_{1:m-1}$. However, one disadvantage of HAPPO is that agent policies must adhere to the sequential updating strategy in the permutation, preventing it from being executed in parallel.

### D.2  BYOL-STYLE AUXILIARY OBJECTIVE

BYOL (Grill et al., 2020) is a powerful self-supervised representation learning algorithm that enforces the similarity of representations of the same visual observation across varied data augmentation. Figure 11 shows the pipeline. BYOL has an online branch and a momentum branch. The momentum branch is utilized to compute a stable goal for learning representations (Chen et al., 2020; He et al., 2019). BYOL is made up of an online encoder $f$, a momentum encoder $\bar{f}$, an online projection head g, a momentum projection head $g$, and a prediction head $q$. The momentum encoder and projection head have the same design as the equivalent online networks and are updated by an exponential moving average (EMA) of the online weights (see Equation 3 in the main manuscript). The prediction head is only employed in the online branch, making BYOL's design asymmetric. BYOL initially generates two views $v$ and $v'$ from an image $x$ using image augmentations. The online branch outputs a representation $y = f(v)$ and a projection $z = g(y)$, and the momentum branch outputs $y' = \bar{f}(v')$ and a momentum version of projection $z' = \bar{g}(y')$. BYOL then uses a prediction head $q$ to fit $z'$ from $z$, i.e., $q(z) \to z'$. Finally, BYOL minimizes the similarity loss between $q(z)$ and a stop-gradient target $sg(z')$.

$$\mathcal{L}^{BYOL}_{f,g,q} = \|q(z) - sg\left(z'\right)\|_2^2 = 2 - 2 \frac{q(z)}{\|q(z)\|_2} \frac{sg\left(z'\right)}{\|sg\left(z'\right)\|_2}.$$

Recent research brings BYOL-style learning objectives to vision-based RL for learning effective state representations and demonstrates promising performance (Hansen & Wang, 2021; Schwarzer et al., 2021a; Yu et al., 2021b; 2022; Yarats et al., 2021), inspired by the success of BYOL in learning visual representations. Future state prediction (Schwarzer et al., 2021a), cycle-consistent dynamics prediction (Yu et al., 2021b), prototype representation learning (Yarats et al., 2021), and invariant representation learning (Hansen & Wang, 2021) are examples of BYOL-style learning. These studies also suggest that supervising/regularizing anticipated representations in the BYOL's projected latent space is more successful than in the representation of original pixel space. Furthermore, BYOL-style auxiliary losses are often trained with data augmentation since it may easily yield two BYOL views. SPR Schwarzer et al. (2021a) and PlayVirtual (Yu et al., 2021b), for example, use random crop and random intensity to enter observations in Atari games. The proposed MAJOR can also be classified as BYOL-style auxiliary objectives.

## D.3 THE TRANSFORMER MODEL AND THE ATTENTION MECHANISM

Transformer (Vaswani et al., 2017) was created originally for machine translation jobs (e.g., input English, output French). It has an encoder-decoder structure in which the encoder maps an input sequence of tokens to latent representations and then the decoder generates a sequence of desired outputs in an auto-regressive manner, with the Transformer taking all previously generated tokens as input at each step of inference. The scaled dot-product attention, which captures the interrelationship of input sequences, is a critical component of the Transformer. The attention function is written as Attention $(\mathbf{Q}, \mathbf{K}, \mathbf{V}) = \text{softmax}\left(\frac{\mathbf{Q}\mathbf{K}^T}{\sqrt{d_k}}\right)\mathbf{V}$, where the $\mathbf{Q}, \mathbf{K}, \mathbf{V}$ corresponds to the vector of queries, keys and values, which can be learned during training, and the $d_k$ represent the dimension of $\mathbf{Q}$ and $\mathbf{K}$. Self-attentions refer to cases when $\mathbf{Q}, \mathbf{K}, \mathbf{V}$ share the same set of parameters.

In practice, the MAT attention mechanism encodes observations and actions using a weight matrix generated by multiplying the embedded queries, as well as keys. The embedded values are multiplied by the weight matrix to output representations. While the encoder's unmasked attention employs a complete weight matrix to extract the interrelationships between agents, the decoder's masked attentions capture sequential actions with triangular matrices. With the properly masked attention mechanism, the decoder can safely output the policy agent-by-agent. We recommend readers give a look at Figure 2, and Algorithm 1 in Wen et al. (2022) to get more details about MAT.

## E   DYNAMIC PROCESS AND SOURCE CODE OF MAJOR

Please refer to https://anonymous.4open.science/r/MAJOR

## F   ADDITIONAL HYPER-PARAMETER IN MAJOR

Table 1: Hyperparameters used for MAJOR.

| Hyperparameter | Value |
|---|---|
| Number of prediction steps $K$ | 1 |
| $\mathcal{L}^{\text{MAJOR}}$ update frequency $H$ | 1 |
| Auxiliary batch size for MAJOR $B'$ | 128 |
| Weight for MAJOR loss $\lambda$ | 1 |
| Hidden units in projection/prediction head | 512 |
| Encoder MEA $\tau$ | 0.01 SMAC and Google Football |
| | 0.05 MA MoJoCo and MAQC |
| EMA update frequency | 1 |
| Number of blocks for joint transition model | 1 |
| Number of heads for joint transition model | 1 |

## G   HYPER-PARAMETER SETTINGS FOR EXPERIMENTS

During experiments, the implementations of baseline methods are consistent with their official repositories, all hyper-parameters left unchanged at the origin best-performing status. The hyper-parameters adopted for different algorithms and tasks are listed in Table 2-8. In particular, the *ppo epochs* and *ppo clip* across different SMAC scenarios are unified to 10 and 0.05 respectively.

Table 2: Common hyper-parameters used for MAJOR, MAT, MAPPO and HAPPO in the SMAC domain.

| hyper-parameters | value | hyper-parameters | value | hyper-parameters | value |
|---|---|---|---|---|---|
| critic lr | 5e-4 | actor lr | 5e-4 | use gae | True |
| gain | 0.01 | optim eps | 1e-5 | batch size | 3200 |
| training threads | 16 | num mini-batch | 1 | rollout threads | 32 |
| entropy coef | 0.01 | max grad norm | 10 | episode length | 100 |
| optimizer | Adam | hidden layer dim | 64 | use huber loss | True |

Table 3: Different hyper-parameters used for MAJOR and MAT in the SMAC domain.

| Maps | ppo epochs | ppo clip | num blocks | num heads | stacked frames | steps | $\gamma$ |
|---|---|---|---|---|---|---|---|
| 27m vs 30m | 5 | 0.2 | 1 | 1 | 1 | 1e7 | 0.99 |
| MMM2 | 5 | 0.05 | 1 | 1 | 1 | 1e7 | 0.99 |
| 6h vs 8z | 15 | 0.05 | 1 | 1 | 1 | 1e7 | 0.99 |

Table 4: Different hyper-parameters used for MAPPO and HAPPO in the SMAC domain.

| Maps | ppo epochs | ppo clip | hidden leyer | stacked frames | network | steps | $\gamma_{MAPPO}$ | $\gamma_{HAPPO}$ |
|---|---|---|---|---|---|---|---|---|
| 27m vs 30m | 5 | 0.2 | 2 | 1 | rnn | 1e7 | 0.99 | 0.95 |
| MMM2 | 5 | 0.2 | 2 | 1 | rnn | 1e7 | 0.99 | 0.95 |
| 6h vs 8z | 5 | 0.2 | 2 | 1 | mlp | 1e7 | 0.99 | 0.95 |

Table 5: Common hyper-parameters used for all methods in the Multi-Agent MuJoCo and Multi-Agent Quadcopter Control domain.

| hyper-parameters | value | hyper-parameters | value | hyper-parameters | value |
|---|---|---|---|---|---|
| gamma | 0.99 | steps | 1e7 | stacked frames | 1 |
| gain | 0.01 | optim eps | 1e-5 | batch size | 4000 |
| training threads | 16 | num mini-batch | 40 | rollout threads | 40 |
| entropy coef | 0.001 | max grad norm | 0.5 | episode length | 100 |
| optimizer | Adam | hidden layer dim | 64 | use huber loss | True |

Table 6: Different hyper-parameter used in the Multi-Agent MuJoCo and Multi-Agent Quadcopter Control domain.

| Maps | MAJOR | MAT | MAPPO | HAPPO |
|---|---|---|---|---|
| critic lr | 5e-5 | 5e-5 | 5e-3 | 5e-3 |
| actor lr | 5e-5 | 5e-5 | 5e-6 | 5e-6 |
| ppo epochs | 10 | 10 | 5 | 5 |
| ppo clip | 0.05 | 0.05 | 0.2 | 0.2 |
| num hidden layer | / | / | 2 | 2 |
| num blocks | 1 | 1 | / | / |
| num head | 1 | 1 | / | / |

Table 7: Common hyper-parameters used for all methods in the Google Research Football domain.

| hyper-parameters | value | hyper-parameters | value | hyper-parameters | value |
|---|---|---|---|---|---|
| critic lr | 5e-4 | actor lr | 5e-4 | gamma | 0.99 |
| gain | 0.01 | optim eps | 1e-5 | batch size | 4000 |
| training threads | 16 | num mini-batch | 1 | rollout threads | 20 |
| entropy coef | 0.01 | max grad norm | 0.5 | episode length | 200 |
| optimizer | Adam | hidden layer dim | 64 | stacked frames | 1 |

Table 8: Different hyper-parameter used in the Google Research Football domain.

| Maps | MAJOR | MAT | MAPPO | HAPPO |
|---|---|---|---|---|
| ppo epochs | 10 | 10 | 5 | 5 |
| ppo clip | 0.05 | 0.05 | 0.2 | 0.2 |
| num hidden layer | / | / | 2 | 2 |
| num blocks | 1 | 1 | / | / |
| num head | 1 | 1 | / | / |

