# OpenReview forum: "Joint-Predictive Representations for Multi-Agent Reinforcement Learning"
_ICLR.cc/2023/Conference — Submitted to ICLR 2023_

### Official Review · Reviewer_hLyZ · 2022-10-22

**Confidence:** 3
**Correctness:** 3
**Technical Novelty And Significance:** 2
**Empirical Novelty And Significance:** 3
**Recommendation:** 6

**Clarity, Quality, Novelty And Reproducibility:**

Clarity: The paper uses somewhat unnecessarily complex exposition to put across a relatively simple point, hindering clarity at times.

Quality: The work is solid and builds on prior work. Presentation could be improved in places (e.g. it's not clear why the Meetup task is included)

Novelty: The work is a simple combination of ideas from different sub-fields (MAT + SSL).

Reproducibility: Source code, hyperparameters, and pseudo code are all provided, so reproducibility should be straightforward.

**Strength And Weaknesses:**

# Strengths
* The main idea is straightforward and improves results.
* The proposed approach is well motivated.

# Weaknesses
* The writing could be more clear.
* Improvements over MAT (which MAJOR builds on) are marginal in many environments.
* Baselines outside of MAT are methods which learn decentralized policies which seems an unfair comparison (unless my interpretation of the proposed approach as being fully centralized is incorrect - see questions).

# Questions
* From the description of the multi-agent transformer, it seems that observations from *all* agents are passed into a single network which computes actions. Given this description, the approach would be fully centralized, correct?
* The authors state that the proposed framework could be applied to CTDE methods, though it is not clear to me how that is the case since the representation learning objective requires sharing information across agents.. Can the authors provide an example of how this may be accomplished in practice?

**Summary Of The Paper:**

This paper presents a self-supervised learning approach for MARL which builds on multi-agent transformers to improve results in several domains. This approach consists of learning a centralized world model as an auxiliary task for learning better representations.

**Summary Of The Review:**

I like the fact that the proposed approach is simple, builds on existing work without adding an overly complicated architecture, and is effective. My main hesitation with the paper is that it overcomplicates the exposition of a relatively simple idea and the presentation in the experimental section could use some work. However, these concerns are relatively minor.

---

> ### Author Response · Authors · 2022-11-13
> **Response to Reviewer hLyZ**
>
> We thank Reviewer hLyZ for the detailed and insightful feedback. We have taken all the comments into consideration and summarized the responses as follows:
>
> > The writing could be more clear.
>
> We are very sorry for our incorrect writing, and we have modified the grammar issues proposed by the reviewers in the revised manuscript.
>
> ---
>
> > Improvements over MAT (which MAJOR builds on) are marginal in many environments.
>
> We would like to refer to what we introduced in response to Reviewer patt and yv1U. For one thing, our method can significantly improve vision-based MARL benchmarks whose observations contain many noisy and irrelevant signals. As for state-based environments, the natural gradient backpropagation of RL loss can make neural network capture good enough information from clean observation vectors. So our proposed auxiliary task is less effective. For another thing, the ablation study of $K$ illustrates that it could act better by adding the forwarding steps of the joint-transition model. The updated results can be found in **Figure 3** of the revised paper.
>
> ---
>
> > Baselines outside of MAT are methods which learn decentralized policies which seems an unfair comparison.
> From the description of the multi-agent transformer, it seems that observations from *all* agents are passed into a single network which computes actions. Given this description, the approach would be fully centralized, correct?
>
> We thank the reviewer for pointing out this issue. There is no doubt that MAT follows the CTCE(centralized training and centralized execution) pattern. But MAT only takes partial observations rather than a global state in the training and execution phases, while CTDE approaches must depend on the global state. However, in many vision-based scenarios, the system state is enormous compared with observations since it has to use an extensive view to cover all agents' views to reflect the system(but most of its messages may be unnecessary). Therefore, the huge-volume state may become the bottleneck of all CTDE MARL methods. Because MAJOR is more effective when applied to vision-based environments, comparing MAT with typical CTDE methods will not result in the mentioned unfair issue.
>
> ---
>
> > The authors state that the proposed framework could be applied to CTDE methods, though it is not clear to me how that is the case since the representation learning objective requires sharing information across agents. Can the authors provide an example of how this may be accomplished in practice?
>
> As we mentioned in response to Reviewer wdZd, we can transfer the core idea of MAJOR into MAPPO and HAPPO with some adaption. The proposed auxiliary learning objective can at least bring improvement for the actor network and can also be helpful for the critic network when using a transformer-liked critic network. More detials can be found in **Appendix C**.
>
> In this work, we attempt to extend MARL with SSL to improve its sample efficiency, which is inspired by the success of incorporating SSL in single-agent RL. In addition, with the development of the MARL benchmarks, the environment's complexity is becoming more complex to match real-world settings. And it's necessary to pay more attention to information comprehension of the various input signals in MARL. Thus, we endeavor to explore the idea of adding SSL priors as an auxiliary task in model-free cooperative MARL algorithms. And we believe that MAT is a better starting point for developing representation learning techniques.

---

> > ### Comment · Reviewer_hLyZ · 2022-11-30
> > **Thank you**
> >
> > Thanks to the authors for their response. My concerns are mostly resolved, though I still feel that comparing to CTDE methods is slightly unfair since the agents trained by those methods are unable to communicate during execution. Furthermore, the authors response seems to indicate that CTDE methods suffer from an additional disadvantage (using the large system state rather than partial observations). It's not clear how this fact addresses the issue of unfair comparison. If anything it seems that this would be an additional factor in making the comparison unfair (especially since these CTDE methods can just as easily use the set of all partial observations rather than the global state).
> >
> > These aren't serious issues since the authors also make a fair comparison to another CTCE method, but are perhaps worth mentioning in the text as a caveat.

---

### Official Review · Reviewer_wdZd · 2022-10-25

**Confidence:** 3
**Correctness:** 3
**Technical Novelty And Significance:** 3
**Empirical Novelty And Significance:** 3
**Recommendation:** 6

**Clarity, Quality, Novelty And Reproducibility:**

* The paper is generally well-written and of good quality, except for the weaknesses listed above.

* The method and empirical analysis is novel.

* There is no reproducibility statement or promise to publish code, and I could imagine it being difficult to reproduce method/results even with provided detailed explanation of components.

**Strength And Weaknesses:**

Strengths:

* The problem is well-motivated; there is a lack of MARL approaches that take advantage of SSL, and there are a number of challenges associated with doing so.

* The proposed objective is a very natural incorporation of a BYOL-like SSL objective into an existing MARL approach. Moreover, the approach appears to be relatively applicable to other model-free MARL methods.

* The writing is generally clear and gets the main ideas and implementation details across. Figure 1 (the illustration of the full MAJOR method) is also a very useful figure for understanding the gist of the method.

* The method shows improvements in sample efficiency (and in some, final convergence in alloted training time) across the board on numerous and diverse multi-agent environments.

Weaknesses:

* Given that other baselines are used apart from MAT in the evaluation section, and that the authors proclaim the approach is “a plug-and-play module”, it would have been really nice to see the loss incorporated into all of the baselines to really drive home this point. Without this, it is less convincing that the module is indeed as simple as “plug-and-play”.

* The methodology section is quite dense, and very few clarifying breathers are given while the stream of details are being revealed. This required me to jump back and forth quite a bit between the more conceptual explanations and the more technical explanations.

* A lot of the intuition/understanding may require familiarity with previous works such as MAT and especially BYOL.

* The results are nice, but the performance improvements (even in terms of sample complexity) are quite modest given the involvement of the approach. Especially for this reason it would have, again, been nice to have seen this module applied to multiple approaches.

* Some grammatical issues / typos (there are others too—I suggest a careful rereading at the sentence-level):

  * First sentence of 3.1 is very hard to parse (run-on sentence and probably grammatically incorrect): “We start with the intuition that encourages the observations and actions representations integrating other agents’ information and embodying agent-level relationships and interactions should reduce the non-stationary during the learning process, thus improving the data efficiency of MARL”

  * Second paragraph 3.1 “since this way can execute” (not proper grammar)

  * First sentence of 3.2 doesn’t make sense: “Multi-Agent Joint-Predictive Representations(MAJOR) is an auxiliary objective to promote the learned representations from the latent space of the sequential observation and action embeddings” (promote what in the learned representations?)

  * Grammar and spelling at top of experiments: “We consider wide-rage MARL benchmarks for evaluating MAJOR and compared MARL algorithms” (”wide-rage” and “and compared”; maybe something like “a wide ranging set of” and “in comparison to relevant SOTA” is meant?).

  * Confusing phrase at the end of “Multi-Agent Quadcopter Control” paragraph in 5.2: “our proposed representation learning framework shows its strength if the underlying approaches don’t work at all” (unclear what is meant here).

**Summary Of The Paper:**

This paper proposes to integrate an SSL auxiliary objective with standard model-free MARL methods (specifically MAT in this paper) to reap the same benefits as CV and single-agent RL has seen with these types of approaches. Because each agent in a typical MARL setup receives a partial observation of the world, in order to apply the SSL auxiliary objective, they treat the individual observations as a sequence of masked views of the global state. Concretely, they use a joint transition model that inputs the individual agent observations and outputs the future observation representation for each agent, which implicitly requires a reconstruction of the global state. The joint-predictive auxiliary objective they introduce on top of MAT is BYOL-like prediction loss. The authors note that a similar objective can be applied to other model-free MARL methods. The new approach, MAJOR, is evaluated on a number of vision- and state-based multi-agent settings. Performance gains in sample efficiency are reported over MAT and other model-free MARL methods across the various settings.

**Summary Of The Review:**

* Overall the paper is well-written and provides a meaningful contribution to the field of MARL. There are some confusing and grammatically incorrect sentences here-and-there that should be corrected, and the claim of the module being plug-and-play is unsubstantiated (and if it were to be substantiated it would make up for the modest improvements in most of the tasks), but it is likely this paper will be useful to the research community in its present state.

---

> ### Author Response · Authors · 2022-11-13
> **Response to Reviewer wdZd (Part 1/2)**
>
> We thank the reviewers for acknowledging the strong performance of this work and the quality of the presentation. We address the comments as follows.
>
> > Given that other baselines are used apart from MAT in the evaluation section, and that the authors proclaim the approach is “a plug-and-play module”, it would have been really nice to see the loss incorporated into all of the baselines to really drive home this point. Without this, it is less convincing that the module is indeed as simple as “plug-and-play”.
>
> Special thanks to you for your good comments. Our proposed auxiliary task for effective representation learning in MARL is built on MAT. As stated in the original paper, the benefit of combining MAT with MAJOR is that it can employ both representations generated from MAT's encoder and decoder. Thus the gradient derived from our auxiliary learning objective can be back-propagated to both the encoder and decoder simultaneously. Contrary to MAT, common-used MARL algorithms(e.g., MAPPO, HAPPO) under the CTDE framework usually take the global state as the input for the critic network, so we need to modify the pipeline of the proposed auxiliary task.
>
>
> **Figure 9** in **Appendix C** demonstrates the combined version of MAPPO. Since MAPPO's critic network doesn't use individual observations, we can only incorporate MAJOR in the actor and make the gradient of auxiliary loss proportionate through the actor network. We are supposing the architecture of the actor network contains more than two MLPs after the encoder. We are supposing the architecture of the actor network contains more than two MLPs after the encoder. In other words, the pipeline of the actor network is "obs→CNN/MLP(parameterized by $\theta$)→obs_rep→MLP(parameterized by $\phi$)→act_rep→MLP(parameterized by $\omega$)→action logits".
>
> Thus given the intermediate outputs of the actor network, we can obtain the observation and action representations. The subsequent pipeline is similar to **Figure 1** in **Section 3**. The difference between 'MAPPO+MAJOR' and 'MAT+MAJOR' are:
>
> 1. We treat MAPPO's actor network as the sequential combination of observation encoder(dubbed as *ObsEncoder*) and action encoder(dubbed *ActEncoder*) followed by the policy head. Alternatively, you can use a separate network as ActEncoder outside the actor network.
> 2. The actor's ActEncoder only takes the observations' representations as input.
> 3. We feed action representations generated from the actor's ActEncoder rather than the executed actions in sampled transitions into the joint-transition model.
>
>
> Note that recently there has been some literature([1], [2]) utilizing Transformer models as the feature extractor for single observations, which contain a sequence of various entities. So applying MAJOR on MAPPO/HAPPO will be similar to the shown pipeline in **Figure 1**. In this situation, we treat the global state as a sequence of observations or elements in the system. And the joint-transition model accepts representations both from actor and critic networks.
>
> We have also evaluated the performance of the modified version of MAJOR in the four-drone LeaderFollower of MAQC. The result can be found in **Figure 10** of **Appendix C**, which illustrates that MAJOR can also improve both MAPPO and HAPPO. But the improvement is not significant because MAJOR can only help the actor network to learn more impact observation representations. Still, it doesn't benefit to critic network, which makes no improvements for value estimation.
>
> [1] Hu, Siyi, Fengda Zhu, Xiaojun Chang and Xiaodan Liang. “UPDeT: Universal Multi-agent RL via Policy Decoupling with Transformers.” *ICLR* (2021).
>
> [2] Wang, Minrui, Ming Feng, Wen-gang Zhou and Houqiang Li. "Stabilizing Voltage in Power Distribution Networks via Multi-Agent Reinforcement Learning with Transformer." *Proceedings of the 28th ACM SIGKDD* (2022).
>
> ---

---

> ### Author Response · Authors · 2022-11-13
> **Response to Reviewer wdZd (Part 2/2)**
>
> > The methodology section is quite dense, and very few clarifying breathers are given while the stream of details are being revealed. This required me to jump back and forth quite a bit between the more conceptual explanations and the more technical explanations.
>
> Thanks for your comment. We will rewritte the methodology section to make it more readable and easy to follow.
>
> ---
>
> > A lot of the intuition/understanding may require familiarity with previous works such as MAT and especially BYOL.
>
> Special thanks to you for your good comments. We have incorporated more details about MAT and BYOL in **Appendix D**, including the introduction of existing methods in MARL, BYOL-style auxiliary objective, and the Transformer model, etc.
>
> ---
>
> > The results are nice, but the performance improvements (even in terms of sample complexity) are quite modest given the involvement of the approach. Especially for this reason it would have, again, been nice to have seen this module applied to multiple approaches.
>
> As shown in response to Reviewer yv1U, we have provided more details about the ablation study in **Appendix B**. The results demonstrate that our method can perform better by appropriately increasing $K$. We have updated the curves in **Figure 3** of **Section 5** in the revised paper. In addition, we applied MAJOR on multiple approaches and displayed the experimental result in **Figure 9** of **Appendix C**.
>
> > Some grammatical issues / typos (there are others too—I suggest a careful rereading at the sentence-level)
>
> We are very sorry for our incorrect writing. According to your advice, we amended the relevant part of the manuscript.

---

> > ### Comment · Reviewer_wdZd · 2022-12-02
> > **Thank you**
> >
> > I thank the authors for addressing my questions thoroughly. My concerns are mostly resolved and I maintain my positive view of the paper.

---

### Official Review · Reviewer_yv1U · 2022-10-25

**Confidence:** 3
**Correctness:** 4
**Technical Novelty And Significance:** 3
**Empirical Novelty And Significance:** 2
**Recommendation:** 6

**Clarity, Quality, Novelty And Reproducibility:**

The paper describes its background and proposal clearly. The quality is high. The experiments are comprehensive, and the results look sound. In addition, experiment settings are described clearly to a certain extent.

**Strength And Weaknesses:**

Strength:
-The empirical result solidly shows the performance of the proposed method.
-The proposal is based on the recent progress and success in SSL, and is insightful.

Weakness:
-The qualitative evaluation and discussion are limited.

**Summary Of The Paper:**

The paper describes Multi-Agent Joint-Predictive representations (MAJOR), which is a self-supervised learning mechanism that allows the MARL system to learn policies in a data-efficient manner.
In MAJOR, observations obtained by individual agents are treated as a masked sequence for representation learning, i.e., masked prediction.
The learned representation should be jointly temporally predictive and consistent across different views overall agents.
The performance  is evaluated through both visionbased
and state-based cooperative MARL benchmarks.
The empirical results show that MAJOR outperforms pre-existing methods.

**Summary Of The Review:**

The proposed method is new and improves the performance of MARL.
Though further discussion is expected, the paper has sufficient strength.
The proposal is insightful and interesting.

---

> ### Author Response · Authors · 2022-11-13
> **Response to Reviewer yv1U**
>
> We greatly appreciate Reviewer yv1U for the positive comments and constructive suggestions. Below we address Reviewer yv1U's suggestions in detail.
>
> > The qualitative evaluation and discussion are limited.
>
> Thank you for your suggestions. In the revised version, we have added the following analysis, discussion, and ablation studies about our proposed method from various perspectives.
>
> - More discussion about the motivation and insights about MAJOR
>
>     In this work, we attempt to extend MARL with SSL to improve its sample efficiency, which is inspired by the success of incorporating SSL in SARL. In addition, with the development of the MARL benchmarks, the environment's complexity is becoming more complex to match real-world settings. And it's necessary to pay more attention to information comprehension of the various input signals in MARL. Thus, we endeavor to explore the idea of adding SSL priors as an auxiliary task in model-free cooperative MARL algorithms. And we believe that our work can give the community more inspirations.
>
> - Discussion about how MAJOR works on MARL
>     In **Figure 5** and **Figure 6** of **Appendix B.1**, we display the training curve of MAJOR loss and V loss in four-drone Flock of MAQC and give the qualitative analysis of MAJOR. The MAJOR loss reduces rapidly, meaning that the MAT's encoder and the joint-transition model can account for other agents' information with the help of the designed auxiliary task. Meanwhile, the global value function approximation also benefits from the compact representation of observations as training progresses since the MAJOR's value loss is lower than MAT's. The more accurately the value function fits, the better the policy optimization effect. And we would like to infer that the non-convergence policy causes the rapid overfitting issue of MAJOR's loss at the beginning of the training phase. Similar results can be found in other scenarios.
>
> - Ablation study of key hyperparameters
>
>     Here we give more details about the ablation study. In our method, the critical hyperparameters are predefined $K$ and $H$, meaning the number of forward steps for the joint transition model and the updating frequency for MAJOR's learning objective, respectively. **Figure 7** in **Appendix B.2** shows the results of the $K$ at $\{1,2,4,8\}$ in four-drone Flock and LeaderFollower of MAQC. We find that extended dynamics modeling consistently improves performance up to roughly. A larger $K$ (e.g., $8$) does not bring further improvement since the network cannot predict long-term future representations as the accumulation of single-step prediction. And **Figure 8** in **Appendix B.2** illustrates that MAJOR's updating frequency can also influence the efficiency of MAT. In our opinion, lazy updating for MAJOR will make representations mismatch with the corresponding policy gradient, thus bringing a negative impact on MAT.

---

> > ### Comment · Reviewer_yv1U · 2022-12-02
> > **Reply**
> >
> > Thank you very much for your revision. I think the quality of the paper is improved.

---

### Official Review · Reviewer_patt · 2022-10-25

**Confidence:** 4
**Correctness:** 3
**Technical Novelty And Significance:** 2
**Empirical Novelty And Significance:** 2
**Recommendation:** 5

**Clarity, Quality, Novelty And Reproducibility:**

 * Most parts of the paper are clear but the writing could be further improved.
 * Although, the code is not attached in the supplementary material, the author provides detailed parameter settings in the Appendix. I think the results could be reproduced according to these detailed settings.

**Strength And Weaknesses:**

**Strengths**:
   * Studying how to learning efficient state abstraction in MARL is an interesting and open problem.
   * The paper conducts extensive experiments on wide-range MARL environments.

**Weaknesses**:
  * The performance improvement of MAJOR is minor compared with the baselines. The training curves of different algorithms are overlapped and thus the improvements are not significant.
  * The benefit of applying SSL to vision-based single-agent RL algorithms is that the policy is learned in the transition-irrelevance or policy irrelevance low-dimensional latent space rather than the original high-dimensional visual space. The paper simply applies SSL to existing MARL methods and shows the experimental results. But the motivation is not very clear. Why applying SSL to MARL is beneficial especially for state-based inputs?
  * Predicting future observations simply based on all agent's current observations may also be inaccurate. In POMDP, all agent's observations may not contain all the information in the state due to the partial observation issue.
  * Minor:
    * "And we define the following mean squared error between the normalized predictions and target projections". 'mean squared error' should be 'cosine similarities'.

**Summary Of The Paper:**

This paper proposes Multi-Agent Joint-Predictive Representations (MAJOR), which applys the self-supervised learning (SSL) technique to MARL, trying to improve the learning efficiency. Specifically, MAJOR builds a transformer-based transition model, which takes all agents observations and actions as inputs and predicts the observations of the next $K$ steps in the latent space. Experimental results on four MARL benchmarks show the SSL auxiliary task could slightly improve the performance of the underlying algorithms.



**Summary Of The Review:**

This paper simply applies SSL to existing MARL methods and shows the experimental results. But the motivation is not very clear. Besides, the experimental results are not significant and the writing of the paper could also be improved. So, in its current form, the reviewer holds the point that the paper is below the acceptance threshold.

---

> ### Author Response · Authors · 2022-11-13
> **Response to Reviewer patt (Part 1/3)**
>
> Thank you for the thoughtful and constructive suggestions! We have considered all the comments and summarized the responses below:
>
> > The performance improvement of MAJOR is minor compared with the baselines. The training curves of different algorithms are overlapped, and thus the improvements are not significant.
>
> The reviewer might have overlooked Figure 4, demonstrating MAJOR's performance in state-based MARL benchmarks. The overlapped training curves result in mainly two reasons.
>
> On one hand, the state-based observations in these environments usually contain the key attributes which demonstrate the position, velocity, and hp of agents, meaning that there are no irrelevance or noisy messages in the vector. Thus the neural network can capture helpful information easily, and selected baselines will get strong performance close to the upper bound with the help of a powerful policy optimization mechanism. So our proposed representation learning task can only take slight improvements w.r.t the original baseline. In other words, for state-based observations, their representations are more effective than vision-based observations through the learning process of MARL methods.
>
> On the other hand, in contrast to low-dimensional state-based vector, learning representations from high-dimensional image-based observations only through policy optimization are more complicated. And our experimental results of MAQC show a significant improvement compared with baselines, which provides the benefit of extra representation learning in current MARL methods for vision-based environments.
>
> In this work, we mainly focus on extending SSL for MARL to improve efficiency, especially in vision-based multi-agent environments. And the results of wide-range benchmarks can demonstrate the generalization of our proposed method for both vision- and state-based scenarios. As for the vision-based benchmark used in the manuscript, Multi-Agent Quadcopter Control, we updated the results of MAJOR by adding the number of prediction steps $K$, we can get the significant performance improvement in both four-drone and eight-drone scenarios. The updated results are shown in **Figure 3** of our revised manuscript.
>
> ---
>
> > The benefit of applying SSL to vision-based single-agent RL algorithms is that the policy is learned in the transition-irrelevance or policy irrelevance low-dimensional latent space rather than the original high-dimensional visual space. The paper simply applies SSL to existing MARL methods and shows the experimental results. But the motivation is not very clear. Why applying SSL to MARL is beneficial especially for state-based inputs?
>
> The reviewer has raised a crucial point; however, we believe that our original writing was unclear and caused the reviewer's confusion.
>
> Firstly, we would like to emphasize that the goal of incorporating SSL in SARL method is to learn more effective representations when fixing the encoder and the number of dimensions of latent space, either visual inputs or state-based inputs. The more informative and rich representations contained in a latent variable, the more effective it is to perform policy optimization. However, in practice, vision-based inputs usually include many noisy and irrelevant components rather than state-based inputs, which brings more obstacles to capturing informative features. Meanwhile, state-based inputs are often designed with critical elements reflecting the physical attributes of the agent with only a few distractions. For example, when we train the same RL algorithm (e.g., SAC) on MuJoCo, it is accessible to extract features from the physical vector(i.e., velocity and position, etc.) via MLP encoder and get high scores, but it is more challenge to use the CNN encoder to obtain good representations and achieve the same performance in DeepMind Control Suite(i.e., visual MuJoCo). Thus we believe that the benefit of applying SSL to both single- and multi-agent RL is more visible on high-dimensional visual space inputs.

---

> ### Author Response · Authors · 2022-11-13
> **Response to Reviewer patt (Part 2/3)**
>
>
> Secondly, as stated in the original paper, we would like to clarify that our work is not simply applied SSL to existing MARL methods. It is worth noting that the representation learning challenge under the MARL context is more difficult than the SARL situation since local observations will be affected by the change in the ego agent's policy and other agents' behavior. As a result, simply applying SSL priors for each agent may be failed due to imperfect information. Furthermore, we believe that our work, which concerned embodying the interaction and fusion among agents in the environment rather than temporal representations for each agent, is not simply to apply SSL to MARL. And our work is motivated by incorporating agent-level interactions to make representations that can take the other agents into account to be more effective. Because of the time limitation of the rebuttal, we will attempt to apply the common-used SSL technique in SARL, named SPR, on MAT to verify whether simply doing the combination will work.
>
> Thirdly, we believe that the vision-based multi-agent system will be paid more and more attention in the future days. It is no doubt that real-world scenarios, such as 2D/3D robotics and eSports games, are almost accepting visual signals but not state-based vectors. Recently, many works are proposed in the community. [1] and MQAC present simulators for controlling drones to visual signals collected with drone cameras. [2] presents a bimanual dexterous manipulation benchmark (Bi-DexHands) according to the cognitive science literature for comprehensive reinforcement learning research. A similar job that predates [2] is OpenAI's human-like robot hand [3], which solves Rubik's Cube and whose input is generated by built-in sensors. Moreover, many robotic navigation benchmarks are presented with more complex visual inputs, such as [4], [5], [6], etc. Meanwhile, in eSports, the famous OpenAI Five and AlphaStar also use images as inputs. These developments indicate that more and more research will be studied on multi-agent environments based on visual inputs. In other words, these 2D/3D robotics and eSports areas provide powerful platforms for researchers to solve real-world problems using MARL. And research based on these areas will also have greater potential to be applied to actual industrial scenarios.However, the current research on MARL seems to be lagging behind. The observations of the common-used benchmarks(e.g. SMAC, Google Research Football, MAMuJoCo) are all based on hand-designed physical values. These environments seem too simplistic compared with the real-world applications mentioned above. So that it is meaningful to make endeavor to explore the idea of representation learning in MARL for vision-based multi-agent systems rather than toy environments. And we believe our work will be useful to the community.
>
> Last but not least, our current presentation might be unclear and misled the reviewer with the confusion of "applying SSL to MARL is beneficial especially for state-based inputs."  As mentioned above, our work is more effective and crucial for MARL problems solving based on visual signals. In the original paper, we stated that our work is built on the latent space, which is general to both vision- and state-based observations. And then, there is a lack of discussion about the benefits of applying MAJOR in various settings. Instead, we only showed the experimental results, and the higher ratio of state-based environments caused the reviewer's confusion and concerns. To address that, we have rewritten the introduction and methodology sections to align with the comments. We hope our discussion and edited sections clarify our motivation and intuition.
>
> [1] Shah, Shital, et al. "Airsim: High-fidelity visual and physical simulation for autonomous vehicles." *Field and service robotics*. Springer, Cham, 2018.
>
> [2] Chen, Yuanpei, et al. "Towards human-level bimanual dexterous manipulation with reinforcement learning." *arXiv preprint arXiv:2206.08686* (2022).
>
> [3] Akkaya, Ilge, et al. "Solving rubik's cube with a robot hand." *arXiv preprint arXiv:1910.07113* (2019).
>
> [4] Deitke, Matt, et al. "Robothor: An open simulation-to-real embodied ai platform." *Proceedings of the IEEE/CVF conference on computer vision and pattern recognition*. 2020.
>
> [5] Deitke, Matt, et al. "Procthor: Large-scale embodied ai using procedural generation." arXiv preprint arXiv:2206.06994 (2022).
>
> [6] Ahn, Michael, et al. "Do as i can, not as i say: Grounding language in robotic affordances." *arXiv preprint arXiv:2204.01691* (2022).
>
> ---

---

> ### Author Response · Authors · 2022-11-13
> **Response to Reviewer patt (Part 3/3)**
>
> > Predicting future observations simply based on all agent's current observations may also be inaccurate. In POMDP, all agent's observations may not contain all the information in the state due to the partial observation issue.
>
>
> We are grateful that the reviewer pointed out this issue. It may not work without any assumptions in Dec-POMDP with imperfect observation mapping functions. In this work, we build MAJOR on MAT, and MAT utilities the local observations to estimate the expected team return under a cooperative MARL context. The MAT's encoder with the value head is used to approximate the global value function and get a good performance, illustrating that it can recover the state and then estimate the team return from the sequence of partial observations. Thus we can assume that all agent's observations in the evaluated Dec-POMDP benchmarks can also contain all the state information, leading to our work becoming practical. At the same time, in our experimental results, we find that MAJOR loss in all experiments reduced rapidly, meaning that our method can work on common-used Dec-POMDP benchmarks without pointed concerns.
>
> As mentioned in the related work section of the original paper, [1] takes a study of learning a dynamic model under model-based MARL in Dec-POMDP, which is similar to the pointed issue. The work proposed a graph-based version adaptation of Predictive State Representations(PSR, [2]) to infer the predictive representations and uses the representations as the input of a policy optimization algorithm. Even though the method is not directly comparable to our model-free way with the auxiliary task, [1] has implicitly proven the feasibility of predicting future representations from individual historical observation. We recommend that the reviewer read the [1] for more information.
>
>
> Furthermore, we believe that adding historical trajectories of agents can address the issue as much as possible. And we will involve both temporal information and agent-level interactions simultaneously in MAT and MAJOR to make them more powerful in our future work.
>
> [1] Zhi Zhang, Zhuoran Yang, Han Liu, Pratap Tokekar, and Furong Huang. Reinforcement learning under a multi-agent predictive state representation model: Method and theory. In International Conference on Learning Representations, 2022.
>
> [2] Michael L Littman, Richard S Sutton, and Satinder P Singh. Predictive representations of state. In NIPS, volume 14, pp. 30, 2001.
>
> ---
>
> > Minor issues & The writing of the paper could also be improved.
>
> We will revise the paper crystal clear to show our motivation, method's details, and discussion about the concerns pointed out by the reviewer.

---

> > ### Comment · Reviewer_patt · 2022-12-05
> > **Response to authors**
> >
> > I appreciate the authors for their response. The authors' response has addressed some of my concerns. I have raised my score to 5.
> >
> > But I still have the following concerns:
> >
> > (1) The major concern is the applicability of the method. As MAJOR is tightly binded with MAT (i.e., requiring the shared encoder and decoder of MAT) in the current version of the paper, the sentence that "the proposed framework is a plug-and-play module for almost common-used MARL methods" is over-claiming. Results with more experiments, i.e., applying MAJOR to typical MARL methods like VDN, QMIX or QPLEX are needed to support the claims. Besides, the performance improvements of applying MAJOR to MAPPO and HAPPO shown in Figure 10 of the Appendix are not significant.
> >
> > (2) MAJOR uses too many Transformer blocks which results in a complex and much bigger network. If the performance improvement is marginal, people will not willing to use such a complex method. Besides, one more baseline is needed to verify the effectiveness of the proposed joint transition model: replacing the Transformer blocks used in the joint transition model with a simple MLP. As the observation encoder and the action decoder are all Transformers and the learned latent embeddings already take the behavior influences of the other agents into consideration, using a simple MLP (which only takes each agent's <$\left.\hat{o}_t^{i}, \hat{a}_t^{i}\right.$> as input) to predict the next observation may be enough.
> >
> > (3) I still hold the point that the empirical results are not significant enough. The performance improvements of MAJOR in Figure 3 (LeaderFollower N=8) and Figure 4 (StarCraftII Multi-Agent Challenge and Google Research Football) are not significant. Besides, for SMAC, why only pick these 3 scenarios as there are many other hard and super hard scenarios?

---

> > > ### Author Response · Authors · 2022-12-10
> > > **Response to Reviewer patt**
> > >
> > > > Concern for the applicability of MAJOR.
> > >
> > > 1. As mentioned in our introduction, we choose MAT, i.e., the SOTA method in cooperative MARL, as the baseline to maximize MAJOR's strength due to its powerful network. Furthermore, we get visible improvement in the visual MARL benchmark. As for the applicability of MAJOR, it is necessary to note that the core idea of MAJOR is to leverage a latent-variable sequence of agents' observations and actions to build contrastive learning objectives and make representations more effective. We propose a joint transition model and a momentum encoder to construct two views of positive version representations and make them similar in latent space. In practice, we can modify the formulation of inputs, e.g., replacing observations with state, to match the baseline algorithm. We can also make the gradient backpropagated into actor and critic submodules. Following the MAJOR's design direction, we believe that our current version of "MAJOR+MAPPO/HAPPO/QMIX" is still used observations and has the potential to be better.
> > > 2. Due to time limitations, we only run the experiments for MAJOR+QMIX on the three super-hard SMAC scenarios across five random seeds. We list the "battle-winning rate(%)" in the following table.
> > >
> > > |Scenario & Method||1m|2m|3m|4m|5m|6m|7m|8m|9m|10m|
> > > |-|-|-|-|-|-|-|-|-|-|-|-|
> > > |6h_vs_8z|QMIX+MAJOR|0±0|±0|0±0|0±0|**1.36±0.64**|**4.92±2.40**|**9.95±3.20**|**18.48±3.06**|**22.6±4.12**|**30.34±7.18**|
> > > ||QMIX|0±0|±0|0±0|0±0|0.78±0.34|3.12±1.12|6.76±1.10|9.02±2.15|13.57±3.78|22.53±15.35|
> > > |MMM2|QMIX+MAJOR|**22.46±12.89**|**57.66±13.95**|**86.12±5.27**|89.74±8.14|93.05±3.08|94.82±1.99|**96.66±1.58**|97.58±1.36|95.9±2.51|95.53±2.38|
> > > ||QMIX|1.97±1.26|30.03±11.3|75.06±9.90|**93.74±1.88**|**94.77±1.38**|**95.03±3.45**|95.44±1.44|**95.69±1.26**|**97.78±0.89**|**96.68±2.38**|
> > > |27m_vs_30m|QMIX+MAJOR|37.88±10.52|70.19±4.21|85.91±4.91|**91.33±5.71**|**90.78±3.87**|**93.59±4.46**|**93.68±3.15**|92.96±3.40|**93.42±3.10**|**94.76±1.96**|
> > > ||QMIX|**44.9±6.04**|**74.78±8.49**|**86.25±3.72**|88.10±4.03|88.36±4.544|91.17±4.03|90.94±3.92|**94.18±2.38**|92.50±2.47|93.86±3.17|
> > >
> > >
> > > > Concern for complex joint transition model.
> > >
> > > We have run the baseline suggested by the reviewer, i.e., replacing the Transformer-based joint transition model(dubbed JMT) to evaluate if we have to use an extra model to take account of other agents' behavior. We selected MAQC scenarios as the testing environment, and the result is shown in the following table. We speculate that the two Transformer-based models implement different agent-level interactions. MAT's encoder is designed to approximate the global value function (i.e., team return) but not the joint transition function. So we cannot predict future representations depending on MAT's encoder, leading to a similar performance between MAT and MLP-JMT-based MAJOR.
> > >
> > > |Scenario & Method||100k|200k|300k|400k|500k|600k|700k|800k|900k|1m|
> > > |-|-|-|-|-|-|-|-|-|-|-|-|
> > > |Flock(N=4)|MAT+MAJOR(Transformer JTM, $K=1$)|**-175.85±10.15**|**-147.28±37.02**|**-118.73±50.22**|**-95.91±29.09**|**-82.84±5.93**|**-81.44±7.27**|**-84.34±3.17**|**-81.35±10.91**|**-77.24±11.62**|**-74.29±12.78**|
> > > ||MAT+MAJOR(MLP JMT, $K=1$)|-183.86±17.05|-154.56±27.03|-158.13±48.39|-149.27±17.79|-155.91±14.82|-157.16±18.91|-143.23±22.94|-152.78±28.09|-143.24±22.94|-135.87±19.08|
> > > ||MAT|-184.39±19.12|-175.30±20.24|-161.49±32.79|-132.85±58.66|-122.34±58.71|-106.99±53.04|-109.71±54.3|-112.92±62.96|-103.53±42.11|-79.67±22.57|
> > > |LeaderFollower(N=4)|MAT+MAJOR(Transformer JTM, $K=1$)|**-31.93±4.82**|**-24.13±8.99**|**-20.31±3.55**|**-18.76±2.80**|**-18.78±3.49**|**-13.96±1.7**|**-14.04±3.02**|**-9.93±1.58**|**-9.83±0.59**|**-9.16±1.50**|
> > > ||MAT+MAJOR(MLP JMT, $K=1$)|-35.72±5.04|-27.28±3,.02|-27.58±3.94|-26.26±4.28|-23.88±4.935|-21.93±3.51|-19.83±2.63|-18.87±2.45|-15.91±0.31|-16.16±0.29|
> > > ||MAT|-31.47±8.01|-27.51±15.82|-29.01±12.19|-29.39±12.66|-26.92±12.06|-23.18±10.44|-18.35±5.89|-19.22±5.80|-17.32±7.85|-20.30±7.33|
> > >
> > >
> > > > The empirical results are not significant enough.
> > >
> > > In this paper, we take several steps to provide an positive answer to the research question "How to incorporate SSL technique in MARL methods?". Our goal is to enhance MARL studies with powerful SSL priors.
> > > As mentioned before, MAJOR in state-based environments(e.g., SMAC, MA MuJoCo, and Google Research Football) will not outperform MAT very much in each scenario since MLP can learn effective representations from physical inputs. Moreover, MAJOR is more suitable for visual environments. We will search for more visual MARL benchmarks and evaluate MAJOR in the future.
> > > Besides, the currently proposed MARL approaches can achieve a 100% winning rate for SMAC in all easy and hard mode scenarios. In other words, it is not necessary to apply SSL on MARL methods in these scenarios to evaluate the performance, so we follow MAT and test MAJOR on three super-hard scenarios, which are evaluated in MAT paper's experiment part.

---

### Comment · Area_Chair_tr9K · 2022-11-24
**Reviewers - pls discuss!**

Dear reviewers,

We are now approaching the end of the discussion period and so far nobody has engaged in discussion with the authors. The authors both updated the paper and provided detailed responses to the reviews. Please reply, clarifying whether your concerns were addressed and if not, why not. Also, if your concerns were indeed addressed either update your score or clearly state why you still believe the score is appropriate.

Many thanks for making this conference a success and for taking your role seriously.

AC

---

### Comment · Area_Chair_tr9K · 2022-12-05
**A few time-criticial questions**

Dear authors,
Can you please specify:
1) how the values for K were chosen in final experiments?
2) Which numbers in the experimental section for baseline performance are directly comparable / validated from 3rd party published results? I am concerned about having experimental results that only compare to internal baselines
3) I don't see any confidence intervals in the paper at all. Which results are *statistically* significant?

Many thanks

AC

---

> ### Author Response · Authors · 2022-12-10
> **Response to AC**
>
> We sincerely thank  AC for your time and efforts. We have considered your comments and summarized the responses below:
>
> > how the values for K were chosen in final experiments?
>
> In MAQC, we display the results of $K=4$ in Figure 3 in Sec.5. Other environments we display the results of $K=1$. And we have also added the ablation study of $K$ in Appendix B.2.
>
> We are very sorry that we only updated the results in one test environment. As mentioned before, visual scenarios are better for evaluating MAJOR's improvement, so we opted to add extra experiments in MAQC under the time limitation of the rebuttal.
>
> > Which numbers in the experimental section for baseline performance are directly comparable / validated from 3rd party published results?
>
> On the one hand, the results in non-vision MARL benchmarks for MAT, HAPPO, and MAPPO are comparable to the number shown in their papers.
>
> On the other hand, according to the relevant literature we can research so far, we have tried our best but have yet to find similar work and experimental results. In other words, we cannot give a 3rd baseline to compare. Moreover, we have also not found visual benchmarks similar to MAQC, which can be formulated as Dec-POMDP. Unfortunately, MAQC's paper did not provide experimental results for common-used MARL methods, i.e., MAPPO, QMIX, HAPPO, etc., so we can only implement internal baselines ourselves.
>
> In our opinion, our method is a first attempt to improve the learning effectiveness of MARL methods by incorporating an auxiliary task. We believe that MAJOR will provide a novel perspective for MARL studies. In addition, we take several steps to provide an positive answer to the research question "How to incorporate SSL technique in MARL methods?".
>
> > I don't see any confidence intervals in the paper at all. Which results are *statistically* significant?
>
> For experiments, we follow what MAPPO/HAPPO/MAT do, run MAJOR with five random seeds for each scenario, and display the mean and variance value for these five runs. The shadows in the pictures demonstrate the variance value, and the lines demonstrate the mean value.

---

### Decision · Program_Chairs · 2023-01-20

**Decision:**

Reject

**Justification For Why Not Higher Score:**

As mentioned above, the experimental results where not convincing in the current version of the paper and the method seems unnecessarily complicated given the limited gains. I also posted a last minute request for clarification to the authors based on the AC meeting and did not obtain a satisfactory answer.


**Justification For Why Not Lower Score:**

NA

**Metareview: Summary, Strengths And Weaknesses:**

This paper proposes a new state encoder / auxiliary task for multi-agent learning in partially observable environments. Since this is a borderline paper the AC and reviewers discussed the contributions in an online meeting and arrived at the following conclusions:
1) The paper is interesting since it presents a potentially innovative solution
2) However, the method is also extremely complicated and it's unclear based on the experimental results whether the complexity is indeed resulting in better performance. For example, the additional results for QMIX vs QMIX + MAJOR do not look significant based on the standard deviation / mean.
3) The final, major concern is the selection of the value "k" for the final run and the lack of significance analysis in all of the results.

The last few points were raised in the online meeting and then posed as questions to the authors. Unfortunately the response was not satisfactory - in particular the authors failed to clarify why a given value of "k" was chosen. Picking this best one a posteriori can be an easy source of bias.


**Summary Of Ac-Reviewer Meeting:**

Since this is a borderline paper the AC and reviewers discussed the contributions in an online meeting and arrived at the following conclusions:
1) The paper is interesting since it presents a potentially innovative solution
2) However, the method is also extremely complicated and it's unclear based on the experimental results whether the complexity is indeed resulting in better performance. For example, the additional results for QMIX vs QMIX + MAJOR do not look significant based on the standard deviation / mean.
3) The final, major concern is the selection of the value "k" for the final run and the lack of significance analysis in all of the results.